# MiGrATe: Mixed-Policy GRPO for Adaptation at Test-Time

## Abstract

Large language models (LLMs) are increasingly being applied to black-box optimization tasks, from program synthesis to molecule design. Prior work typically leverages in-context learning to iteratively guide the model towards better solutions. Such methods, however, often struggle to balance exploration of new solution spaces with exploitation of high-reward ones. Recently, test-time training (TTT) with synthetic data has shown promise in improving solution quality. However, the need for hand-crafted training data tailored to each task limits feasibility and scalability across domains. To address this problem, we introduce MiGrATe—a method for *online* TTT that uses GRPO as a *search* algorithm to adapt LLMs at inference without requiring external training data. MiGrATe operates via a mixed-policy group construction procedure that combines on-policy sampling with two off-policy data selection techniques: greedy sampling, which selects top-performing past completions, and neighborhood sampling (NS), which generates completions structurally similar to high-reward ones. Together, these components bias the policy gradient towards exploiting promising regions in the solution space, while preserving exploration through on-policy sampling. We evaluate MiGrATe on four challenging domains—word search, molecule optimization, hypothesis+program induction on the Abstraction and Reasoning Corpus (ARC), and natural-language hypothesis search on DiscoveryBench—and find that it consistently outperforms both inference-only and TTT baselines, demonstrating the potential of online TTT as a solution for complex search tasks without curated training data.

## 1 Introduction

Large language models (LLMs) have emerged as general-purpose tools for solving a wide range of black-box optimization problems (Boiko et al., 2023; Ramos et al., 2023; Liu et al., 2024). These models offer a flexible interface for generating candidate solutions, both in structured tasks, e.g., molecule design (Ranković & Schwaller, 2023; Kristiadi et al., 2024; Gruver et al., 2024), and unstructured, natural-language tasks, e.g., scientific hypothesis generation (Lu et al., 2024; Majumder et al., 2025; Agarwal et al., 2025b).

Recent work has shown that in-context learning (ICL) (Brown et al., 2020) can effectively be used to steer LLMs toward higher-quality outputs in such tasks (Meyerson et al., 2023; Yang et al., 2024b; Agarwal et al., 2025a). However, ICL alone lacks a principled mechanism to balance *exploration* of novel solution areas with *exploitation* of known high-reward ones (Krishnamurthy et al., 2024) based on simply injecting a history of candidates in-context. Without this balance, the model may either get trapped in local optima or waste sampling budget on unpromising regions of the solution space.

To improve LLM-based search, recent methods have explored *test-time training* (TTT) (Sun et al., 2020; Hardt & Sun, 2024)—a paradigm inspired from the human ability to generalize from a few examples (Yu et al., 2025a), in which the LLM is adapted at inference time for a specific problem instance before sampling a set of candidate solutions to evaluate. Similarly, some works have explored the use of off-policy reinforcement learning to efficiently learn suitable sampling distributions (Levine et al., 2020; Yan et al., 2025). However, these approaches either rely on carefully hand-crafted, task-specific data generation strategies or assume availability of expert demonstration

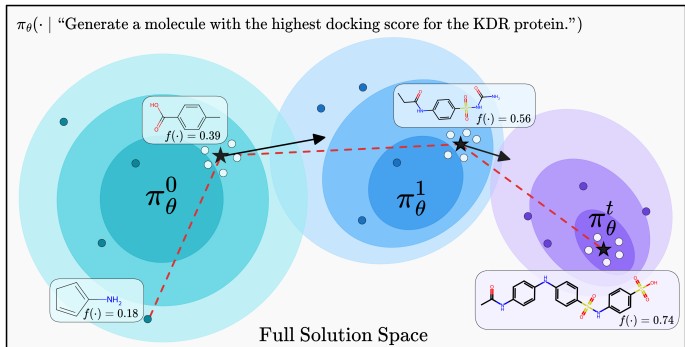

Figure 1: **Overview of MIGRATE.** Given a search problem, MIGRATE iteratively searches for optimal solutions by sampling candidates and updating its policy model $\pi_\theta^t$ using mixed-policy GRPO. In each iteration, we combine online samples (•) from the current policy distribution, top-performing past solutions (⋆) as greedy references, and samples drawn from the neighborhoods of greedy solutions (○) to form a GRPO group. The resulting group is used to update $\pi_\theta^t$ and *migrate* towards a sampling distribution that is likely to generate higher-quality solutions according to $f(\cdot)$.

data (Akyürek et al., 2025; Li et al., 2024), both of which limit the generality and scalability of such solutions.

To address these shortcomings, we cast search as an online reinforcement learning problem and leverage group relative policy optimization (GRPO) (Shao et al., 2024) to iteratively find promising regions of the search space, balancing exploration and exploitation. In practice, this means iteratively optimizing a set of LoRA parameters added to a pre-trained LLM in order to improve the instance-specific sampling distribution to generate better solutions. We, thus, propose **MIGRATE** (**Mi**xed-policy **GR**PO for **A**daptation at **T**est-Time), a method for *online* TTT that enables adaptive search with LLMs *without* requiring any external, handcrafted training data. Our method combines:

1. **On-policy sampling**, which ensures continual exploration of the solution space,

2. **Greedy sampling**, which reuses top-performing past completions to exploit known high-reward regions, and

3. **Neighborhood sampling (NS)**, which generates structurally similar variants of high-reward completions to facilitate local exploration.

Crucially, all components in MIGRATE use only model-generated signals, eliminating the need for any external training data. We perform experiments on four challenging domains with diverse solution spaces and reward functions—word search, molecule optimization, hypothesis+program induction using the Abstraction and Reasoning Corpus (ARC) (Chollet, 2019), and data-driven discovery using DiscoveryBench (Majumder et al., 2025). Across all domains, we find that MIGRATE outperforms both inference-only and TTT baselines, demonstrating the effectiveness of lightweight parameter updates, using online TTT with mixed-policy guidance, in providing a generic approach to LLM-based black-box optimization.

To summarize, our main contributions are as follows:

- We introduce MIGRATE, a method to search for optimal solutions with LLMs using an online test-time training (TTT) algorithm without external demonstrations.

- We propose a mixed-policy group construction strategy that combines on-policy sampling with two novel off-policy techniques—greedy sampling and neighborhood sampling.

- We conduct comprehensive experiments across four diverse domains, showing that MIGRATE outperforms both inference-only and TTT baselines in complex black-box optimization tasks.

## 2 RELATED WORK

**Test-time training.**    Test-time training (TTT) aims to improve model performance on distribution shifts by updating models at inference. Sun et al. (2020) introduced TTT using a self-supervised objective on images to adapt network weights at test time. Hardt & Sun (2024) demonstrate that fine-tuning LLMs on data closely related to each test prompt can yield large accuracy gains, extending TTT to reasoning tasks. Hübotter et al. (2025) show that nearest-neighbor retrieval for test-time fine-tuning often wastes effort on redundant examples, and instead propose an active-learning method that chooses maximally informative examples to reduce model uncertainty.

**Local-structure methods.**    Instance-based learning (or "local learning") (Atkeson et al., 1997) is a common framework in machine learning where local structure is exploited around a test point to improve model accuracy, e.g., locally-weighted regression (Cleveland, 1979). In modern practice, this manifests as retrieving nearest-neighbor examples to guide adaptation, referred to as retrieval-augmented generation (RAG) or case-based reasoning (CBR) (Lewis et al., 2020; Das et al., 2021; Thai et al., 2023; Agarwal et al., 2024). In reinforcement learning, local policy search methods (e.g., off-policy local improvements, trust-region updates) behave like hill-climbers in the policy space.

**Evolutionary computation.**    EvoTune (Surina et al., 2025) uses an LLM as a policy-generating operator in an evolutionary loop, then applies RL fine-tuning to iteratively improve it. AlphaEvolve (Novikov et al., 2025) similarly creates an agent that uses multiple LLMs and automated evaluators to propose and refine codebases via an evolutionary framework. FunSearch (Romera-Paredes et al., 2024) pairs a pre-trained LLM with an automated evaluator and repeatedly samples and scores code functions, effectively evolving programs to solve mathematical problems. In these systems, the "population" of programs or policies evolves over generations, often via an islands model or parallel ensembles, to avoid local traps.

**Bayesian optimization and LLMs.**    Bayesian optimization (BO) is an optimization approach that consists of using a surrogate model and an acquisition function in an iterative process to optimize some objective function. Recent works integrate LLMs at various stages of the BO process, leveraging their semantic understanding and ability to encode information. LLAMBO (Liu et al., 2024) uses the natural language capabilities of LLMs to be surrogates for both parts of the BO framework by having it generate and evaluate solution proposals. BOPRO (Agarwal et al., 2025a) embeds solutions into a latent space and employs an acquisition function to adapt the proposal prompt for an LLM, effectively steering the the model towards promising regions in the solution space. InstructZero (Chen et al., 2023) uses BO to learn soft prompts, which are then converted into instruction prompts to elicit better instruction following behavior from LLMs. Our work focuses on optimizing the LLM as a proposal mechanism for generating optimal solutions with respect to a black-box function. Internally, MIGRATE operates an acquisition-like strategy to formulate prompts that evoke higher quality solutions from the LLM.

## 3 BACKGROUND

**GRPO.**    Group relative policy optimization (Shao et al., 2024) is a reinforcement learning algorithm used to fine-tune LLMs that replaces the value function in Proximal Policy Optimization (PPO) training (Schulman et al., 2017) with an estimate derived from Monte Carlo samples instead. In particular, in each iteration of training, GRPO constructs a group $\mathcal{G}$ of $N$ completions, typically sampled from the current model, and calculates the advantage for every completion as a relative comparison to the group. Let $\pi_{\theta_{\text{old}}}$ and $\pi_\theta$ denote the model policies (LLM parameters, in our case) before and after taking a gradient step. Given a task prompt $P_\mathcal{T}$ and a set of completions sampled from the current model $\{o_i : o_i \sim \pi_{\theta_{\text{old}}}\}_{i=1}^N$, the GRPO loss objective is defined as

$$\mathcal{L}_{\text{GRPO}}(\theta) = -\frac{1}{\sum_{i=1}^N |o_i|} \sum_{i=1}^N \sum_{t=1}^{|o_i|} \Big[ \min\big(r_{i,t}(\theta)\hat{A}_{i,t}, \text{clip}(r_{i,t}(\theta), 1 - \varepsilon_{\text{low}}, 1 + \varepsilon_{\text{high}})\hat{A}_{i,t}\big) \Big], \quad (1)$$

$$\text{where} \quad r_{i,t}(\theta) = \frac{\pi_\theta(o_{i,t} \mid P_\mathcal{T}, o_{i,<t})}{\pi_{\theta_{\text{old}}}(o_{i,t} \mid P_\mathcal{T}, o_{i,<t})}, \quad \text{and} \quad \hat{A}_{i,t} = r_i - \text{mean}(\{f(o_i)\}_{i=1}^N)$$

are the policy ratio and advantage estimates, respectively, for each token in each completion, $f(\cdot)$ is a reward function that provides a scalar score for each completion, $\text{clip}(\cdot, \cdot, \cdot)$ is a clipping function to prevent large updates during optimization, and $\varepsilon_{\text{low/high}}$ are clipping hyperparameters.

**On-, off-, and mixed-policy optimization.** Typically, reinforcement learning (including GRPO) operates in an *on-policy* manner, where new solutions are sampled using $\pi_\theta$ (i.e., the policy being trained) to estimate the loss for the next training step. On the other hand, some works have argued that on-policy training may constrain learning to only the capabilities of the base LLM itself, resulting in echo chambers (Zhao et al., 2025; Yue et al., 2025) that prevent novel task generalization. This problem is further exacerbated in the sparse reward scenario, where the base model is unable to generate solutions that elicit non-zero reward, thus leading to degenerate policy gradients. To address this, *off-policy* optimization (Levine et al., 2020) has been proposed as an effective strategy that leverages previously collected expert demonstrations for training instead of online samples. However, a purely offline strategy can result in learning policies that are unable to generalize at inference time (Fujimoto et al., 2019; Kumar et al., 2019). Consequently, recent work (Yan et al., 2025) shows that a combination of online and offline samples, called *mixed-policy* optimization, can outperform either strategy used in isolation.

## 4 MIGRATE: METHODOLOGY

The focus in this work is on finding optimal solutions with respect to a black-box objective function $f(\cdot)$ under a finite sampling budget $B$. To this end, we are interested in using GRPO as a *search* algorithm, wherein a single example query is used as the input for a search task across multiple sampling iterations. The goal, then, is to learn query-specific parameters that shift the model's sampling distribution iteratively, improving the quality of solutions that are generated.[1][2]

**Overcoming sparse rewards in search.** As described earlier, purely on-policy learning is often unable to find an appropriate sampling distribution for a single query within a limited budget due to sparse rewards, i.e., when solutions sampled from the current policy do not result in useful policy gradients to make progress. At the same time, both off- and mixed-policy strategies require access to known expert demonstrations, which we assume are not available in our setting. We, therefore, present **MIGRATE**—a mixed-policy optimization strategy for GRPO that generates off-policy data via (a) selecting high-performing solutions from the model's own sampling history, and (b) sampling variations from the neighborhoods of observed high-performing solutions. In each iteration, MIGRATE "mixes" on- and off-policy samples to construct a group of completions $\mathcal{G}$, which is then used to compute the policy gradient with respect to the loss function in Equation 1. This process is repeated until either the optimal solution is found or the sampling budget is exhausted.

### 4.1 MIXED-POLICY GROUP CONSTRUCTION FOR SEARCH

Given a search task $\mathcal{T}$ and a corresponding task prompt $P_\mathcal{T}$ for the LLM, our goal is to construct a new group $\mathcal{G}_t$ composed of $N$ completions in each search iteration $t$ to compute a policy gradient via GRPO. We introduce two off-policy data selection techniques—**greedy** and **neighborhood sampling (NS)**—which we combine with on-policy sampling to generate test-time training data. Intuitively, both techniques are designed to bias policy gradients to *exploit* known high-quality solutions sampled thus far, while on-policy sampling encourages *exploration*. In experiments (§ 5), we find that the simultaneous application of greedy and NS off-policy data selection (i.e., MIGRATE; Algorithm 1) results in the best performance.

---

[1]This is in contrast to the more typical setting of training a generalizable model with multiple examples. See the appendix for a complete description of modifications we incorporate from previous work beyond the original formulation from Shao et al. (2024).

[2]Note that throughout this work, we use LoRA fine-tuning (Hu et al., 2022) instead of full-model training.

**On-policy sampling.** Let $\alpha\ (\leq N)$ be the number of completions sampled from the current policy model, i.e., at timestep $t$, we generate on-policy completions (or observations) $\mathcal{O}_{\text{online}} := \{o_i : o_i \sim \pi_\theta^{t-1}(\cdot \mid P_\mathcal{T})\}_{i=1}^\alpha$ using temperature-based ancestral sampling.

**Greedy sampling.** Let $\mathcal{D}$ be a database of completions, which may be composed both of any candidate solutions available *a priori* as well as all attempts sampled from the model in previous search iterations. In greedy off-policy data selection, if $\mathcal{D} \neq \emptyset$, we sample $\beta\ (\leq N)$ known completions from $\mathcal{D}$ that are high-quality. In particular, we first greedily select the top-$k$ completions from $\mathcal{D}$ with respect to $f(\cdot)$ and then randomly sample $\beta$ completions from the top-$k$, i.e., $\mathcal{O}_{\text{greedy}} := \{o_i : o_i \sim \text{topk}_f(\mathcal{D})\}_{i=1}^\beta$, where $\text{topk}_f(\mathcal{D})$ returns the best-$k$ completions from $\mathcal{D}$ with respect to $f$.

**Neighborhood sampling.** While greedy sampling explicitly encourages the exploitation of high-quality samples, it limits exploration of the solution space and is prone to optimizing for local optima (Krishnamurthy et al., 2024; Agarwal et al., 2025a). To mitigate this, we incorporate a complementary off-policy sampling strategy grounded in a *continuity assumption*—namely, that small variations in a model's parameter space yield small shifts in the average quality of sampled solutions (see Fig. 2). This assumption motivates exploration within neighborhoods of known high-quality candidates by prompting the model to generate stochastic variations of greedy samples, thereby producing *new* solutions that may both provide useful variations for better policy gradients as well as solutions that may outperform previous samples. In practice, we construct a single neighborhood sampling prompt $P_{\text{NS}}$ composed of $\beta$ greedy samples along with an instruction to generate $\gamma\ (\leq N)$ to construct the NS set of solutions $\mathcal{O}_{\text{NS}} := \{o_i : o_i \sim \pi_\theta^{t-1}(\cdot \mid P_{\text{NS}})\}_{i=1}^\gamma$.

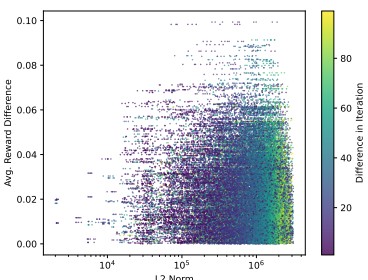

Figure 2: **Visualizing parameter space continuity.** Each point is a pairwise comparison between two sets of LoRA parameters, indicating distance (x-axis) and average difference in sample quality (y-axis), over 100 search iterations on Semantle. Performance converges with a decrease in pairwise distances, whereas at larger distances, performance varies, indicating the variability encountered when exploring.

**MIGRATE.** To balance exploration and exploitation during test-time training with GRPO, MIGRATE integrates both off-policy techniques with on-policy sampling by combining $\mathcal{O}_{\text{online}}$, $\mathcal{O}_{\text{greedy}}$, and $\mathcal{O}_{\text{NS}}$ into a single group $\mathcal{G}_t$, with the constraint that $\alpha + \gamma + \beta = N$ in each iteration [3] (see Algorithm 1). We compute the loss on $\mathcal{G}_t$ with respect to the task prompt $P_\mathcal{T}$, irrespective of how the sample was generated. While on-policy sampling encourages exploration of new solutions, greedy sampling promotes exploitation by reusing high-quality completions from a running database, and neighborhood sampling introduces structured exploration via local variations of the greedy samples. Empirically, we find that this combination produces higher-quality search results than any single strategy alone.

---

**Algorithm 1 Solution search with MIGRATE**

**Input**: Task $\mathcal{T}$, black-box function $f$, budget $B$
**Parameters**: GRPO group size $N$, $\alpha$ on-policy samples, $\beta$ greedy samples, $\gamma$ neighborhood samples
**Output**: Best solution $o_{\text{best}}$

1: **Initialize:** Policy $\pi_\theta^0 \leftarrow$ LLM, task prompt $P_\mathcal{T}$, database $\mathcal{D} \leftarrow \emptyset$, timestep $t \leftarrow 0$, $o_{\text{best}} \leftarrow \emptyset$
2: **while** $|\mathcal{D}| < B$ **do**
3:     $t \leftarrow t + 1$
4:     $\mathcal{O}_{\text{online}} \leftarrow \{o_i : o_i \sim \pi_\theta^{t-1}(\cdot \mid P_\mathcal{T})\}_{i=1}^\alpha$
5:     $\mathcal{O}_{\text{greedy}} \leftarrow \{o_i : o_i \sim \text{topk}_f(\mathcal{D})\}_{i=1}^\beta$
6:     $P_{\text{NS}} \leftarrow$ Build NS prompt using $\mathcal{O}_{\text{greedy}}$
7:     $\mathcal{O}_{\text{NS}} \leftarrow \{o_i : o_i \sim \pi_\theta^{t-1}(\cdot \mid P_{\text{NS}})\}_{i=1}^\gamma$
8:     $\mathcal{G}_t \leftarrow \mathcal{O}_{\text{online}} \oplus \mathcal{O}_{\text{greedy}} \oplus \mathcal{O}_{\text{NS}}$
9:     $\mathcal{D} \leftarrow \mathcal{D} \oplus \mathcal{O}_{\text{online}} \oplus \mathcal{O}_{\text{NS}}$
10:     $o_{\text{best}} \leftarrow \arg\max_{o_i \in \mathcal{D}} f(o_i)$
11:     **if** $o_{\text{best}}$ is optimal **then**
12:         **return** $o_{\text{best}}$
13:     **end if**
14:     $\pi_\theta^t \leftarrow$ Update using GRPO with $\mathcal{G}_t$ (Eq. 1)
15: **end while**
16: **return** $o_{\text{best}}$

---

[3]We keep constant the number of new solutions sampled from the LLM for fair comparison with baselines.

# 5 EXPERIMENTS

## 5.1 SEARCH TASKS

We evaluate MIGRATE by conducting experiments on four text-based search tasks—Semantle (word search), Dockstring (molecule optimization), ARC (hypothesis + program search), and DiscoveryBench (data-driven hypothesis search).

**Semantle.** Semantle (Agarwal et al., 2025a) is a word-search task, where the goal is to identify a held-out English word (e.g., *"polyethylene"*) within a limited number of guesses. The black-box function used indicates how semantically close a guessed word is to the target, which is computed using cosine similarities over SimCSE (Gao et al., 2021) embeddings, following prior work. Each search problem is initialized with a warmstart set of 20 words (randomly sampled from the word2vec index (Mikolov et al., 2013)) and corresponding black-box scores. We conduct evaluation using 10 hidden words and 5 warmstart sets for each of them, resulting in a total of 50 problem instances.

**Dockstring.** García-Ortegón et al. (2022) provides a suite of challenging molecule optimization tasks that reflect real-world problems in drug discovery. We focus on a multi-objective optimization task: generating molecules, represented as SMILES strings (Weininger, 1988), that simultaneously maximize druglikeness and binding affinity, quantified by QED (Bickerton et al., 2012) and negative Vina scores (Trott & Olson, 2010), respectively. We use a scalarized multi-objective black-box function (Equation 2) that places a greater weight on Vina scores than QED, reflecting the common prioritization of binding affinity over druglikeness when evaluating a molecule's drug efficacy (Hughes et al., 2011; Wenlock et al., 2003). Following prior works (Yuksekgonul et al., 2024), we run our evaluation with 58 pharmaceutically-relevant protein targets.

**ARC.** The Abstraction and Reasoning Corpus (ARC) (Chollet, 2019) is a benchmark of 400 grid-based puzzles that involves inferring the transformation logic from a small set of input-output grid pairs and applying it to a held-out test grid. Recent methods improve performance via data augmentation with invertible transformations (Akyürek et al., 2025) or by combining program synthesis with transductive strategies (Li et al., 2024). We take an inductive hypothesis + program search approach (Wang et al., 2024), where natural language transformation algorithms are hypothesized and translated into Python programs. We report two accuracy metrics: *pass@2*, which measures whether any of the top-2 common outputs from the programs that solve the train set matches the test grid, and *oracle*, which provides credit if any of the sampled programs solves the test grid. Note that oracle accuracy reflects a coarse ability to find a distribution that can generate the correct solution. We follow prior work (Agarwal et al., 2025a) and use a Hamming-distance based black-box function. [4]

**DiscoveryBench.** DiscoveryBench (Majumder et al., 2025) is a benchmark to evaluate hypothesis search ability in data-driven scientific discovery. It includes a set of discovery tasks extracted from real-world scientific publications, each represented by a research query and a corresponding dataset, aiming to find statistically verifiable natural-language hypotheses that can answer the given queries. We assume oracle feedback in each iteration to help guide search (akin to feedback from a human researcher) using a scalar score representing the degree to which a generated hypothesis matches the gold hypothesis using a Beta belief distribution elicited from an LLM (Agarwal et al., 2025b). We evaluate performance using both the belief-based black-box function (average belief and % of queries where the belief was maximized) as well as the hypothesis match score (HMS) from Majumder et al. (2025), which provides an LLM-judge evaluation of hypotheses based on contexts, variables, and relationships. Additionally, our analyses found that the HMS tends to score hypotheses with even minor deviation from the gold context as zeros. Therefore, we introduce HMS-$\rho$, a relaxation of HMS that allows an LLM to provide partial scores for the context, i.e., $\{0, 0.5, 1.0\}$ instead of $\{0, 1\}$ only, in order to lend graded improvement information.

---

[4] Due to hardware limitations, we truncate prompts at 2048 tokens in all experiments. As a result, only 200 out of 400 tasks in ARC-Full could be evaluated with their full context.

## 5.2 BASELINES

**Inference-only.** We evaluate five inference-only sampling strategies (Random, NS, OPRO, Evolvution, and BOPRO) for Semantle, Dockstring, and ARC, and use Reflexion (following Majumder et al. (2025)) as the baseline for DiscoveryBench:

- **Random**, which generates completions by sampling from the base model using the task prompt.
- **Neighborhood Sampling (NS)**, which samples completions from a prompt that includes top-performing solutions from previous iterations to encourage local exploration.
- **OPRO** (Yang et al., 2024b), which generates completions using a prompt that builds a trajectory of top-performing solutions as a textual gradient to discover improved solutions.
- **Reflexion** (Shinn et al., 2024), which iteratively improves LLM performance by generating natural-language feedback ("self-reflection") using solutions from past iterations.
- **Evolution**, which iteratively optimizes generated solutions by mutating sampled solutions according to an evolutionary pipeline.[5]
- **BOPRO** (Agarwal et al., 2025a), which uses latent space Bayesian optimization over solution embeddings to search for better sampling distributions via context engineering over past solutions.

**Test-time training.** Beyond inference-only methods, we evaluate three variants of our GRPO-based test-time training (TTT) approach:

- **GRPO** is the base algorithm, using a fixed task prompt and sampling $N$ on-policy completions from the model as it is being trained (i.e., $\alpha = N$, $\beta = 0$, $\gamma = 0$).
- **GRPO-Greedy** augments GRPO by using greedy off-policy sampling to select $\beta$ previous completions to place in the group at each iteration (i.e., $\alpha, \beta > 0$ and $\gamma = 0$).
- **Online DPO** (Guo et al., 2024) samples $N$ on-policy completions in each iteration, which are used to construct preference pairs and calculate a policy gradient using the standard DPO objective (Rafailov et al., 2023).
- **MIGRATE** is our full method, combining on-policy exploration, greedy sampling of top completions, and neighborhood sampling for local exploration (i.e., each of $\alpha, \beta, \gamma > 0$).

We provide complete details of our experimental settings in Appendix A.1, including the values used for $\alpha$, $\beta$, and $\gamma$ for different tasks, and sensitivity analyses of these choices in Appendix B.3.

**Additional baselines.** We also evaluate MIGRATE (OPRO), a variant of MIGRATE that replaces the neighborhood sampling (NS) prompt with the OPRO prompting strategy for local exploration (Appendix B.5), as well as explore an alternative strategy for selecting $\mathcal{O}_{\text{greedy}}$ using an islands-based evolutionary search method (Appendix B.1).

**Models.** Our main results on Semantle and Dockstring are presented using LLaMA-3.2-3B-Instruct (AI@Meta, 2024). For ARC, we use LLaMA-3.1-ARC-Potpourri-Induction-8B (Li et al., 2024), a fine-tuned version of LLaMA-3.1-8B-Instruct (AI@Meta, 2024) trained on synthetic Python programs that solve ARC training tasks. The latter decision is driven by the bespoke nature of the ARC challenge, where base models are entirely unable to generate valid solutions. For DiscoveryBench, we use Qwen2.5-7B-Instruct (Yang et al., 2024a) for generating experiment plans and GPT-5-nano (OpenAI, 2025) for the remainder of the agentic loop (code, reviews, and analyses). We use Qwen2.5-7B-Instruct for belief elicitation during search, but report final accuracy using GPT-4o (as in Majumder et al. (2025)).

## 6 RESULTS AND DISCUSSION

**MIGRATE outperforms both inference-only and TTT baselines.** Across tasks, we run each method until either the correct solution is found or a pre-defined budget of solution candidates

---

[5]We use OpenEvolve for our implementation (Novikov et al., 2025; Sharma, 2025).

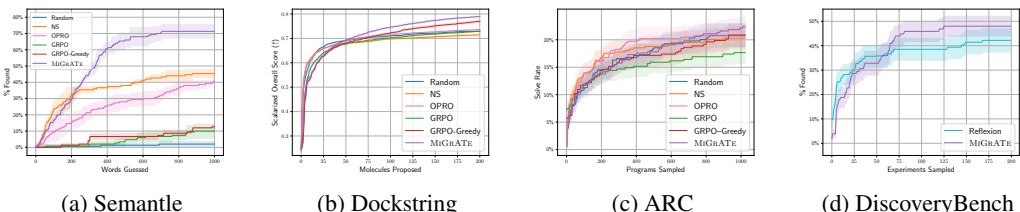

|  (a) Semantle | (b) Dockstring | (c) ARC | (d) DiscoveryBench |

Figure 3: **Best-so-far performance results.** (a) On Semantle, MIGRATE outperforms all baselines, improving the second-best (NS) by $25\%$. (b) In Dockstring, MIGRATE surpasses baselines after 50 proposals. (c) On ARC, MIGRATE solves more tasks than baselines at the full budget. (d) On DiscoveryBench, MIGRATE outperforms Reflexion after 65 experiments.

is proposed and evaluated (1000 for Semantle, 200 for Dockstring, 1024 for ARC, and 200 for DiscoveryBench). We report our results on each search task in Table 1 and provide a best-so-far plot to trace search behavior across sampling budgets in Figure 3. We find that mixed-policy GRPO via MIGRATE outperforms each inference-only baseline and TTT ablation.

On Semantle, MIGRATE outperforms baselines except for BOPRO by $\geq$ 21 percentage points. As shown in Figure 3(a), across the 50 problem instances averaged over 3 repeat runs, MIGRATE surpasses inference-only NS after 200 guesses ($\sim$20 MIGRATE iterations), pointing to the effectiveness of explicit gradient updates in finding high-quality solutions versus in-context optimization alone. BOPRO's better performance suggests that incorporating a BO strategy into MIGRATE to construct the NS prompt could be beneficial.

| | Semantle | Dockstring | | | ARC | |
|---|---|---|---|---|---|---|
| **Method** | **% Found** | **QED ($\uparrow$)** | **Vina Score ($\downarrow$)** | **Overall Score ($\uparrow$)** | **Pass@2 (%)** | **Oracle (%)** |
| Random | $2.00 \pm 1.63$ | $\mathbf{0.91 \pm 0.00}$ | $-9.92 \pm 0.15$ | $0.73 \pm 0.00$ | 20.75 | 28.00 |
| NS | $45.30 \pm 2.49$ | $0.87 \pm 0.01$ | $-9.65 \pm 0.21$ | $0.71 \pm 0.00$ | 20.25 | $\underline{29.50}$ |
| OPRO | $40.70 \pm 1.89$ | $0.90 \pm 0.00$ | $-9.94 \pm 0.06$ | $0.74 \pm 0.00$ | 20.75 | 27.75 |
| Evolution | $49.33 \pm 4.11$ | $0.89 \pm 0.03$ | $-9.56 \pm 0.09$ | $0.72 \pm 0.01$ | - | - |
| BOPRO | $\mathbf{84.67 \pm 0.94}$ | $0.89 \pm 0.00$ | $-10.28 \pm 0.04$ | $0.77 \pm 0.00$ | - | - |
| Online DPO | $4.00 \pm 4.90$ | $0.90 \pm 0.02$ | $-9.41 \pm 0.09$ | $0.71 \pm 0.01$ | - | - |
| GRPO | $10.00 \pm 4.32$ | $\underline{0.91 \pm 0.00}$ | $-10.09 \pm 0.05$ | $0.73 \pm 0.00$ | 17.75 | 27.00 |
| GRPO-Greedy | $12.70 \pm 0.94$ | $\overline{0.90 \pm 0.01}$ | $-10.80 \pm 0.19$ | $0.77 \pm 0.00$ | $\underline{21.00}$ | $\mathbf{30.00}$ |
| MIGRATE | $\underline{71.30 \pm 4.11}$ | $0.90 \pm 0.00$ | $\mathbf{-11.00 \pm 0.07}$ | $\mathbf{0.79 \pm 0.00}$ | $\mathbf{22.25}$ | $\mathbf{30.00}$ |

| | DiscoveryBench | | | | | |
|---|---|---|---|---|---|---|
| **Method** | **Belief** | **% Found (Belief)** | **HMS** | **% Found (HMS)** | **HMS-$\rho$** | **% Found (HMS-$\rho$)** |
| Reflexion | $0.758 \pm 0.022$ | $17.00 \pm 3.78$ | $\mathbf{0.293}$ | $\mathbf{13.00}$ | 7.00 | $\mathbf{0.273}$ |
| MIGRATE | $\mathbf{0.795 \pm 0.018}$ | $\mathbf{20.00 \pm 4.13}$ | 0.285 | 11.00 | $\mathbf{13.00}$ | 0.268 |

Table 1: **Search performance.** Results are averaged over three random seeds for Semantle and Dockstring, with standard deviations reported. For ARC and DiscoveryBench, we report using single runs (due to expense) but report standard deviation via bootstrapping. Top-2 results in each column are marked with bold and underline, respectively. MIGRATE outperforms on all but one metric (QED) on Semantle, Dockstring, and ARC [6]. On DiscoveryBench, MIGRATE finds hypotheses that are more similar to the gold as measured by the belief-based black-box function and HMS-$\rho$, while showing marginally lower performance using HMS.

On Dockstring, Table 1 shows that MIGRATE synthesizes molecules with higher scalarized scores (according to Equation 2), i.e., jointly optimizing for QED and Vina. Further, in Figure 3(b), we see that MIGRATE outperforms all baselines on average after 50 molecule proposals. We also show the search trace of different methods in Figure 4.

On ARC, we report performance over a single run (due to hardware constraints), and report standard deviation via bootstrapping. From Figure 3(c) and Table 1, we find that MIGRATE does outperform

---

[6]Due to hardware limitations, we only evaluated Evolution and BOPRO on a subset of the ARC benchmark in B.5

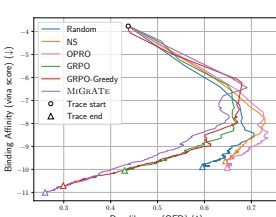 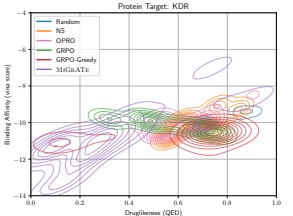 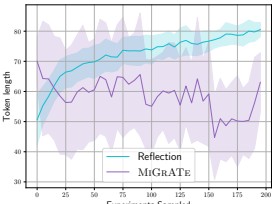

(a) Dockstring search trace    (b) SMILES distribution (KDR)  (c) DiscoveryBench Token Length

Figure 4: **Search behaviors.** (a) Vina and QED scores for best molecules found as search progresses. Each trace starts from 3 fragments (acetamide, pentane, and benzene). (b) Distribution of binding affinity and druglikeness for KDR target. MIGRATE explores a broader region of chemical space, including low-affinity and low-druglikeness. (c) Experiments generated by Reflexion monotonically increase in token length with time, while those by MIGRATE remain stable on average.

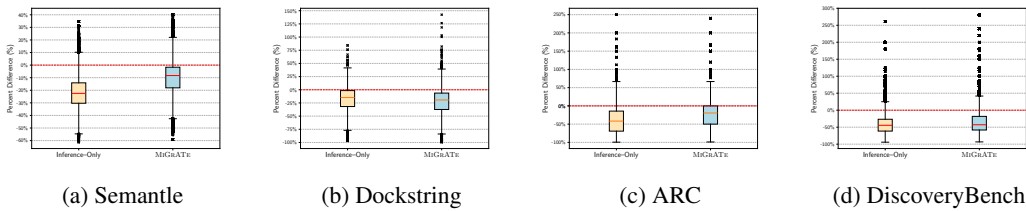

(a) Semantle          (b) Dockstring          (c) ARC          (d) DiscoveryBench

Figure 5: **Relative to the best-so-far.** Percentage difference between samples from MIGRATE (versus inference-only NS) relative to their best-so-far scores during optimization. Across search iterations, MIGRATE generates solutions (a) with higher quality on average (as indicated by the higher mean; except on Dockstring), and (b) those that show greater jumps in performance over the best-so-far (i.e., the outliers), indicating better search and exploration ability.

baselines, though, with modest gains akin to behavior reported by prior work on LLM-based program search. We do, however, find that MIGRATE solves all but two tasks also solved by baselines. We note that MIGRATE also outperformed Evolution and BOPRO on a subset of the ARC benchmark in Appendix B.5.

On DiscoveryBench, we evaluate 100 tasks from the test set, ensuring a balanced distribution of domains and question types. The Reflexion baseline solves 44 queries, while MIGRATE solves 48, crucially, without any natural language feedback. As shown in Figure 3(d), MIGRATE outperforms Reflexion after proposing 65 experiment plans, which corresponds to 13 training iterations.

**TTT methods produce qualitatively different solutions than inference-only methods.** On Semantle, across all runs, we find that MIGRATE is the only method to find all 10 hidden words. Although BOPRO achieves a higher average accuracy, it fails to every find one of the ten hidden words. Furthermore, only MIGRATE and its ablations can optimize for specific words, like "birthstone," demonstrating the ability to navigate the unique search landscape for such terms. On Dockstring, as shown in Figure 4(a), the best-performing SMILES strings found using TTT methods (MIGRATE and its ablations) show a distinct optimization pattern, focusing more on Vina scores than those from inference-only methods. While MIGRATE is capable of generating molecules with high QED scores ($> 0.8$), optimization prefers to reduce QED to below $0.3$ in exchange for better Vina scores. This reflects the multi-objective function in Equation 2, which weighs Vina scores more than QED. On DiscoveryBench, the lengths of experiment plans from Reflexion monotonically increase over time, while plans from MIGRATE remain stable on average (Figure 4(c)). Notably, the best plans are consistently shorter ($<115$ tokens), suggesting MIGRATE is able to prioritize these during search.

**What search behaviors are observed with MIGRATE?** We analyze the quality of samples generated by MIGRATE and NS (inference-only) and compare them in Figure 5. We measure the

relative difference between the scores of each solution and the best-so-far performance when that solution is sampled, then compare the distributions of these differences between the two methods. On Semantle and ARC, MIGRATE demonstrates the ability to improve upon its previously best-found solution in contrast to the behavior seen with the inference-only strategy, which often samples solutions with no improvement. In Dockstring, MIGRATE generates more invalid molecules than inference-only approaches, suggesting broader exploration of the solution space (Figure 4(a) and (b)). Many of the proposed molecules are also longer and more complex SMILES strings, evidenced by a $44\%$ increase in average length. Despite proposing more invalid molecules, MIGRATE still finds molecules that improve upon the best-so-far with larger gains than with inference-only.

## 7 CONCLUSION

We introduced MIGRATE, a method for online test-time training of LLMs that enables efficient search in black-box optimization tasks without requiring handcrafted training data. By leveraging Group Relative Policy Optimization (GRPO) along with a novel mixed-policy group construction strategy—comprising on-policy, greedy, and neighborhood sampling—MIGRATE effectively balances exploration and exploitation. Our experiments across four text-based domains demonstrate the efficacy of MIGRATE to improve LLM-based search. Future work may include scaling online TTT to multi-step decision-making and integrating stronger uncertainty-aware acquisition strategies to further improve sample efficiency.

## 8 REPRODUCIBILITY STATEMENT

We include the source code along with instructions to reproduce our experiments as part of the supplementary material. We also provide the specific hyperparameters used in Appendix A.1.

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

# A APPENDIX A

## A.1 EXPERIMENTAL SETTINGS

**Semantle.** The black-box function we use is the cosine similarity of vector representations generated using the SimCSE Gao et al. (2021) sentence embedding model, where the score for a proposed word $x$ for a hidden target word $y$ is computed by comparing the embeddings for the sequences "What is a $\{x\}$?" and "What is a $\{y\}$?". The number of warmstart candidates is 20. Our main results with NS and MIGRATE selects $\mathcal{O}_{\text{greedy}}$ by uniformly sampling among the top-3 completions found so far according to their black-box scores.

In MIGRATE, we execute GRPO for 100 generation steps where we sample a batch of 10 words in each step for a total sampling budget of 1000 words. In each step, we sort the generated batch of words by their scores and construct a group of 5 completions, each consisting of 2 words each. Each completion is assigned the maximum score of the two words as its reward.

For the Random baseline, we sample 1000 words using the task prompt. For the NS baseline, we sample 10 words using the NS prompt for 100 iterations. Similarly, for the OPRO baseline, we also sample 10 words using the OPRO prompt for 100 iterations. We provide, in-context, the top-10 words found so far for every OPRO-based method.

In our Online DPO baseline, we used the same training hyperparameters as GRPO. In each training iteration, we generate 10 words which equates to 5 preferences. Here, words with the higher score are preferred (ranked) over those with lower scores.

**Dockstring.** The black-box function we use is a linear function of the binding affinity (Vina) and druglikeness (QED). We use RDKit's `MolFromSmiles` to sanitize a given generated SMILES string. If this process fails due to an invalid format structure or molecule, we assign the generated molecule a score of 0. If the molecule is valid, we compute the QED and Vina scores on the given protein target. We then compute the overall score of these two metrics as follows:

$$s_{\text{overall}}(\text{molecule}, \text{protein}) = 1 - \mathcal{N}(\texttt{Vina}(\text{molecule}, \text{protein}) + (1 - \texttt{QED}(\text{molecule})) \quad (2)$$

Where $\mathcal{N}$ denotes min-max normalization to the range [0,1]. The QED score is bounded between 0 and 1, and we assume the Vina score to be between 0 and -13.0 kcal/mol. In practice, the binding affinity is a much higher priority than the druglikeness. Given our equation and the value ranges for computing $s_{\text{overall}}$, our black-fox function accurately emphasizes the Vina score about 10 times more than the QED score.

For the Random baseline, we sample 200 molecules using the task prompt. For the NS baseline, we sample 3 molecules using the task prompt and 2 molecules using the NS prompt in each iteration for 40 iterations. We select $\mathcal{O}_{\text{greedy}}$ from the top-1 molecule found so far in NS and MIGRATE. For the OPRO baseline, we sample 5 molecules using the OPRO prompt for 40 iterations. We provide, in-context, the top-5 molecules proposed so far for every OPRO-based method.

In our Online DPO baseline, we used the same training hyperparameters as GRPO. In each training iteration, we generate 5 molecules and create 10 pairwise preferences. Here, molecules with a higher overall score according to Eq. 2 are preferred (ranked) over those with lower scores.

**ARC.** The black-box function we use is a hamming-distance based metric. We run all input grids with the sampled program and compute the proportion of cells in the ground-truth grid that matches the output grid. We assign a reward of 0 if the program does not terminate within 10 seconds of execution. During training, the reward is given by averaging the score across all training input grids of the given ARC task. If the output grid is larger than the ground-truth, then we assign a score of 0.

For the Random baseline, we sample 1024 programs using the task prompt. For the NS baseline, we sample 12 programs using the task prompt and 4 programs using the NS prompt for 64 iterations. We note that this Random baseline is equivalent to the main evaluations ran by Li et al. Additionally, our TTT baselines on ARC in the inductive setting are not an entirely fair comparison to prior works that do TTT in the transductive setting. We select $\mathcal{O}_{\text{greedy}}$ as the top-1 program found so far for

| Hyperparameter | Value |
|---|---|
| Model | Llama 3.2 3B Instruct Grattafiori et al. (2024) |
| Learning rate | 1e-5 |
| Group size | 5 |
| LoRA rank | 64 |
| LoRA alpha | 16 |
| Training steps | 100 |
| Iterations per step | 2 |
| GRPO $[\alpha, \gamma, \beta]$ | $[5, 0, 0]$ |
| GRPO-Greedy $[\alpha, \gamma, \beta]$ | $[4, 0, 1]$ |
| MIGRATE $[\alpha, \gamma, \beta]$ | $[0, 4, 1]$ |

Table 2: MIGRATE hyperparameters for Semantle

| Hyperparameter | Value |
|---|---|
| Model | Llama 3.2 3B Instruct Grattafiori et al. (2024) |
| Learning rate | 5e-5 |
| Group size | 5 |
| LoRA rank | 64 |
| LoRA alpha | 16 |
| Training steps | 40 |
| Iterations per step | 1 |
| GRPO $[\alpha, \gamma, \beta]$ | $[5, 0, 0]$ |
| GRPO-Greedy $[\alpha, \gamma, \beta]$ | $[4, 0, 1]$ |
| MIGRATE $[\alpha, \gamma, \beta]$ | $[2, 2, 1]$ |

Table 3: MIGRATE hyperparameters for Dockstring

both NS and MIGRATE. Similarly, for the OPRO baseline, we sample 12 programs using the task prompt and 4 programs using the OPRO prompt for 64 iterations. Due to hardware limitations and to maintain a fair comparison with MIGRATE, we only provide one program in-context for the OPRO prompt.

**Discoverybench.** The main black-box function we use is a belief-based score which represents the extent a model believes a generated hypothesis matches the gold hypothesis. In our implementation, we create a Beta belief distribution from 10 samples from a base Qwen 2.5 7B-Instruct model Yang et al. (2024a). We observed that using the Qwen model for this task performed similarly to sampling from GPT-4o OpenAI et al. (2024). During Reflextion and MIGRATE, we perform early stopping once a hypothesis with a belief score greater than 0.8 is found.

For the Reflextion baseline, we perform 40 iterations where we sample 5 experiments in each iteration. We evaluate and generate a reflection for the 5 experiments in each iteration to pass into the next. Similarly, in MIGRATE, we perform 40 training iterations where each iteration generates 5 experiments.

## A.2 GRPO FORMULATION

We remove the KL term in the original GRPO objective. Following DAPO Yu et al. (2025b), we utilize token-level normalization, which assigns more balanced rewards to individually generated tokens—alleviating the bias towards longer responses. We also set $\varepsilon_{\text{low}} = 0.2$ and $\varepsilon_{\text{low}} = 0.28$ which DAPO finds to promote exploration of low-probability tokens that perform well. Dr. GRPO Liu et al. (2025) also divides the sum of loss by a constant instead of the total sequence length

| Hyperparameter | Value |
|---|---|
| Model | BARC Li et al. (2024) |
| Learning rate | 1e-5 |
| Group size | 16 |
| LoRA rank | 128 |
| LoRA alpha | 32 |
| Training steps | 64 |
| Iterations per step | 1 |
| GRPO $[\alpha, \gamma, \beta]$ | $[16, 0, 0]$ |
| GRPO-Greedy $[\alpha, \gamma, \beta]$ | $[15, 0, 1]$ |
| MIGRATE $[\alpha, \gamma, \beta]$ | $[11, 4, 1]$ |

Table 4: MIGRATE hyperparameters for ARC

| Hyperparameter | Value |
|---|---|
| Model | Qwen 2.5 7B Instruct Yang et al. (2024a) |
| Learning rate | 1e-5 |
| Group size | 5 |
| LoRA rank | 128 |
| LoRA alpha | 32 |
| Training steps | 40 |
| Iterations per step | 2 |
| MIGRATE $[\alpha, \gamma, \beta]$ | $[2, 2, 1]$ |

Table 5: MIGRATE hyperparameters for Discoverybench

to completely remove any completion length bias. Although we did not use this formulation in our experiments, there should be no substantial differences since there is not high variability in the solution lengths in the domains we studied. Following Dr. GRPO, we do not scale the advantage by the standard deviation of the group's rewards. By doing so, we avoid biasing weight optimization on groups that perform extremely well or poorly on a given prompt. While our online prompt always remains constant, this bias is relevant for our NS prompt which can vary across iterations.

### A.3 COMPUTATIONAL RESOURCES

All experiments were conducted on a cluster of NVIDIA GPUs. We utilize a mixture of A100 (40GB and 80GB), L40S, and A40 GPUs. TTT methods on ARC-Full were run with A100 (80GB) GPUs due to the higher memory requirements. Our implementation of MIGRATE is based on the TRL 0.19.0 implementation of GRPO from HuggingFace von Werra et al. (2020). We also utilize Unsloth Daniel Han & team (2023) and vLLM Kwon et al. (2023) to enable higher sampling throughput and lower memory usage.

**Runtimes.** The average runtime for MIGRATE on each Semantle problem was 93 seconds on an A100 GPU, while for NS, it is 83 seconds for each problem. On Dockstring, the average runtime across all GPU types on each molecule optimization task was 7.5 minutes for MIGRATE and 8.2 minutes for NS. The average runtime on each ARC task with early stopping is 51 minutes for MIGRATE and 47 minutes for NS on an A100 GPU. The average runtime for on each DiscoveryBench query with early stopping is 61 minutles for MIGRATE and 46.6 minutes for Reflexion.

As seen from these runtimes, test-time training with MIGRATE does not add substantial latency over inference-only methods. Most of the latency can be attributed to routines common to both optimization strategies. For example, in ARC, the primary source of latency is solution (program) sampling, where in Dockstring, the main source is the black-box function, i.e., simulating whether the proposal molecule can dock onto the target protein.

## B  APPENDIX B: ADDITIONAL EXPERIMENTS

### B.1  ISLAND-BASED EVOLUTION ALGORITHM

We implement an island-based evoluationary algorithm as an alterative to top-$k$ for selecting $\mathcal{O}_{\text{greedy}}$. We created a database inspired by Ellenberg et al. (2025) to store generated solutions and sample them for constructing neighborhood sampling. The island model organizes the solutions into isolated islands of solutions that are evolved independently.

At every training step, we iterate to another "island" in the database in a cyclic order. We then sample a solution stored at this island to construct our neighborhood sampling prompt. We note that unlike prior works Ellenberg et al. (2025); Surina et al. (2025) we do not construct additional subclusters of solutions within each island. This was done due to the low sampling constraints of our experiments but can also be seen as using a single cluster per island. Sampling from an island is carried out by an exploitation strategy with probability $p$ and an exploration strategy with probability $1 - p$. With the exploitation strategy, we randomly select a top solutions on the island that is also considered a globally top-$k$ solution across all islands. If the island does not have a solution that is in the top-$k$ solution for all islands then we fall back on the exploration strategy. With the exploration strategy, we randomly select among the top solutions on the island that are *not* one of the globally top-$k$ solutions.

We periodically migrate a percentage of the top-performing solutions from each island to their neighboring islands according to a ring topology. This maintains a balance of exploring diverse solutions in isolation and preventing the algorithm from spending too much time on low-performing solutions.

We conduct a comparison of using NS and MIGRATE with three different strategies for selecting the solution to sample neighbors from: Top-1, Top-3, and Evolution. For each of these configurations we use 10 neighborhood samples, 0 online samples, and 0 greedy samples. Fig. 6 shows that Top-3 outperforms Top-1 and that using our evolution-based strategy outperforms Top-3 in both NS and MIGRATE methods. While Top-3 shows the better initial gains in both NS and MIGRATE, the evolution-based strategy narrowly outperforms it by 1000 samples. Much like our other results in Table. 1, we also observe that the MIGRATE equivalent of each NS variation performs better – reinforcing the pattern that TTT improves search performance.

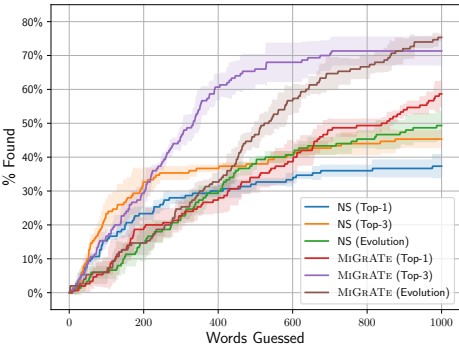

Figure 6: **Comparing selection methods for NS.** Evolution-based selection shows slower initial gains but results in more consistent improvements than using a top-$k$ sampling strategy–resulting in better final performances.

### B.2  CAN RELATED TASKS BOOTSTRAP SEARCH?

We investigate whether fine-tuned weights from TTT can generalize to other tasks. After running MIGRATE on every task, we perform TTT again on unsolved tasks and bootstrap the method with the learned weights of its "nearest" solved task.

In this experiment, we attempt to solve ARC tasks that were not solved by MIGRATE. For each unsolved task, we determine its "nearest" solved task by evaluating this task using the solution

program from every solved task. We pass the training inputs of the unsolved task into each program and determine the nearest solved task to be the one whose solution program achieve the highest reward from our hamming distance-based reward function.

Once the nearest solved task is identified, we use its fine-tuned weights from MIGRATE as the initializing point for solving the unsolved task. This procedure aims to transfer inductive biases that may have been learned from structurally similar tasks, enabling the model to efficiently explore more viable programs on the unsolved task. This tests whether there is an advantage to initializing search via TTT from a more informed starting point on problems where starting with the base model fails.

We see marginal improvements from bootstrapping search with learned weights from MIGRATE. Fig. 7 shows that initializing Random Sampling and MIGRATE with the nearest solved task's weights allowed each respective method to solve tasks that were initially unsolvable by the base model. Notably, bootstrapping Random Sampling with nearest weights was able to solve more tasks than executing MIGRATE on the base model.

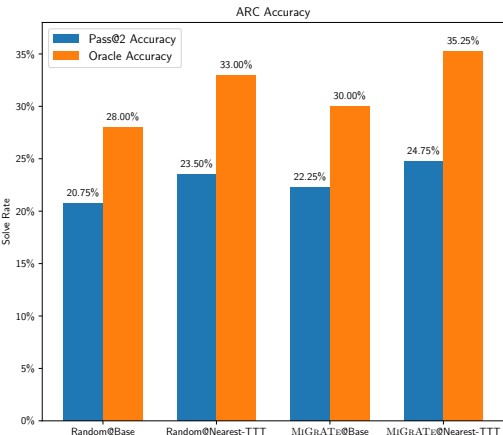

Figure 7: **Bootstrapping with nearest weights on ARC-Full.** Bootstrapping Random and MI-GRATE with initial weights learned from one round of MIGRATE shows slight improvement on total tasks solved.

### B.3 HYPERPARAMETER SENSITIVITY ANALYSES

#### B.3.1 VARYING $\alpha$ AND $\gamma$ SAMPLES

We conduct experiments on Semantle, Dockstring, and ARC-Small to investigate the tradeoff involved in varying the ratio of online to neighborhood samples within a GRPO group in MIGRATE. ARC-Small is a subset consisting of 54 tasks with grids up to a maximum of 64 cells, created to measure variance across search methods via repeat runs.[7]

Throughout these experiments, we fix the number of greedy samples at $\beta = 1$. The results in Fig. 8 reveals that the optimal configuration of online sand NS samples vary across domains. Particularly, Semantle benefits from more NS samples, Dockstring performs the best with an equal ratio of samples, while ARC prefers a higher proportion of online samples. These results highlights the importanced of tuning $\alpha$ and $\gamma$ when applying MIGRATE to different domains.

#### B.3.2 VARYING $\beta$ SAMPLES

We explore varying the number of greedy samples on Semantle. In these experiments, we run MIGRATE with $\alpha = 0$ onlines amples, $\beta$ greedy samples, and $N - \beta$ neighborhood sampless. As shown in Fig. 9a, performance remains relatively similar over $\beta = 0, 1, 5, 10$ with a small trend

---

[7]Note that we ensure ARC-Small maintains the same difficulty distribution as ARC-Full.

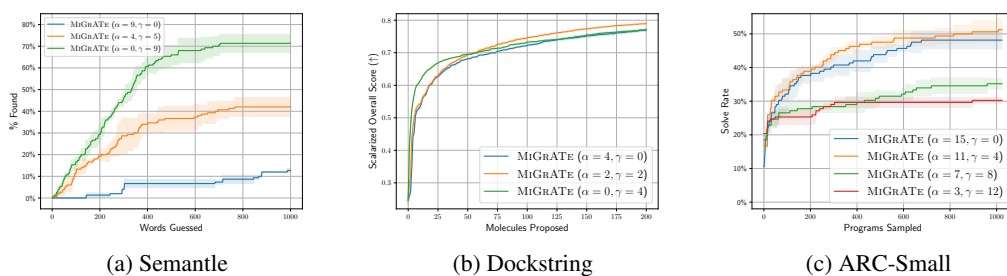

(a) Semantle         (b) Dockstring         (c) ARC-Small

Figure 8: **Varying $\alpha$ and $\gamma$.** We vary the number of online and NS samples per group in MIGRATE. **(a)** On Semantle, we found that the strategy of using no online samples to be the most successful by a significant margin. **(b)** On Dockstring, we found that using only NS samples yield better performances at smaller budgets and a configuration of equal amounts of online and NS samples to achieve the best final performance. **(c)** On ARC-Small, we found the mixed configuration of $\alpha = 11$ and $\gamma = 4$ to perform the best.

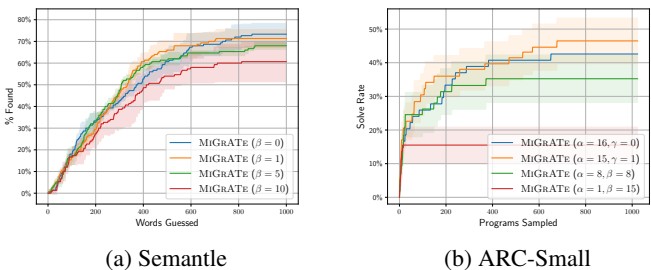

(a) Semantle         (b) ARC-Small

Figure 9: **Comparing $\beta$ on Semantle and ARC.** MIGRATE shows a bias towards smaller $\beta$ for better performance on Semantle and ARC-Small.

of better performance with smaller $\beta$. In tandem with the results on varying $\gamma$, this supports the potential of more off-policy methods of performing TTT with GRPO.

### B.4 VARYING REWARD FUNCTION SPARSITY

To investigate the impact of reward function sparsity on the performance of MIGRATE, we conduct experiments on Semantle and systematically vary the sparsity of the reward signal. Specifically, we modify the reward function such that rewards below a certain threshold are rounded down to zero, thereby introducing sparsity into the reward signal. Let $f(o_i)$ be the original value from a black-box function for a solution $o_i$. We introduce a sparsity threshold $T \in [0, 1]$ and define the modified reward function $\hat{f}(\cdot)$ as follows:

$$\hat{f}(o_i) = \begin{cases} 0 & \text{if} f(o_i) < T \\ f(o_i) & \text{otherwise.} \end{cases} \tag{3}$$

Next, we apply this sparsity function to MIGRATE and OPRO on Semantle to evaluate the effect of sparsity on search performance. We test with $T = [0, 0.25, 0.5, 0.75, 1.0]$. Specifically, $T = 0$ corresponds to the original reward function $f(\cdot)$ and $T = 1.0$ results in a binary reward function where only the oracle solution maps to a non-zero reward.

As expected, in Figure 10(a,b), both MIGRATE and OPRO show a decline in performance as the reward sparsity increases. Interestingly, however, Figure 10(c) demonstrates that MIGRATE shows higher robustness to sparse rewards than the purely in-context OPRO baseline, with the gap between MIGRATE and OPRO progressively increasing with higher sparsity.

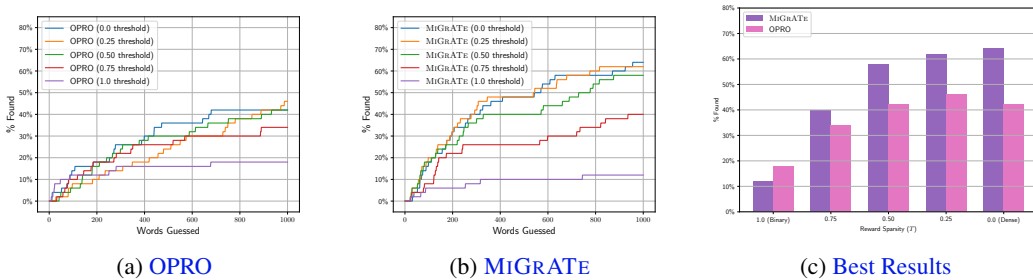

(a) OPRO          (b) MIGRATE          (c) Best Results

Figure 10: **Impact of reward sparsity on MIGRATE and OPRO. (a,b)** MIGRATE and OPRO see similar decreases in performance on Semantle as reward sparsity increases. **(c)** MIGRATE also shows more robustness to the reward sparsity by scaling better to denser rewards than OPRO. Notably, MIGRATE matches the best OPRO performance at the second highest sparsity setting.

| Method | Semantle | Dockstring | | |
| | % Found | QED ($\uparrow$) | Vina Score ($\downarrow$) | Overall Score ($\uparrow$) |
|---|---|---|---|---|
| NS | $45.30 \pm 2.49$ | $0.87 \pm 0.01$ | $-9.65 \pm 0.21$ | $0.71 \pm 0.00$ |
| OPRO | $40.70 \pm 1.89$ | $0.90 \pm 0.00$ | $-9.94 \pm 0.06$ | $0.74 \pm 0.00$ |
| MIGRATE | $\mathbf{71.30 \pm 4.11}$ | $\mathbf{0.90 \pm 0.00}$ | $\mathbf{-11.00 \pm 0.07}$ | $\mathbf{0.79 \pm 0.00}$ |
| MIGRATE (OPRO) | $65.3\% \pm 2.49$ | $\mathbf{0.90 \pm 0.00}$ | $-10.80 \pm 0.10$ | $0.78 \pm 0.00$ |

| Method | ARC-Small | |
| | Pass@2 (%) | Oracle (%) |
|---|---|---|
| NS | $48.15 \pm 0.00$ | $55.56 \pm 1.51$ |
| OPRO | $\mathbf{50.62 \pm 1.75}$ | $\mathbf{59.26 \pm 0.00}$ |
| Evolution | $44.44 \pm 1.51$ | $\underline{57.41 \pm 0.00}$ |
| BOPRO | $22.22 \pm 0.80$ | $22.22 \pm 0.80$ |
| MIGRATE | $\mathbf{51.23 \pm 3.49}$ | $\mathbf{62.35 \pm 0.87}$ |
| MIGRATE ($\gamma$-OPRO) | $44.44\% \pm 3.02$ | $\underline{55.56 \pm 0.04}$ |
| MIGRATE ($\gamma$-Evolution) | $\underline{45.68 \pm 0.01}$ | $\underline{46.30 \pm 0.00}$ |

Table 6: **Comparing alternative sampling strategies.** We compare the inference-only and MIGRATE (TTT) performance of different sampling techniques. All results are averaged over three random seeds, with the standard deviation reported. The best result in each column is marked in bold and the second best result is underlined. Despite OPRO showing better performance over NS when comparing with the inference-only strategy, we see that NS demonstrates higher performance than OPRO when combined with MIGRATE.

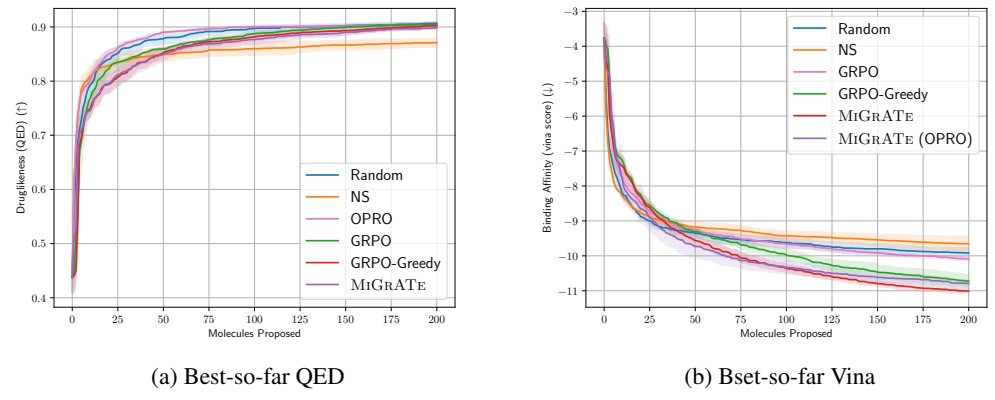

(a) Best-so-far QED          (b) Bset-so-far Vina

Figure 11: **QED and Vina Score plots for Dockstring.**

## B.5 ALTERNATIVE LOCAL STRUCTURE SAMPLING IN MIGRATE?

We experiment with the alternative of using OPRO in place of neighborhood sampling (NS) in MI-GRATE. Our results in Table. 6 show similar results between MIGRATE and MIGRATE (OPRO) on Dockstring and more favorable results towards MIGRATE on Semantle and ARC-Small. Compared to other baselines in Table 1, MIGRATE (OPRO) only underperforms relative to MIGRATE on Semantle and Dockstring. Notably, on ARC-Small, incorporating TTT into OPRO substantially degrades performance compared to inference-only OPRO. We also observe that OPRO achieves better performance than NS across most metrics. The varying performance of MIGRATE (OPRO) across domains suggests that NS is more compatible than OPRO with MIGRATE. In addition, the greater improvement achieved by using NS over OPRO suggests that the NS strategy of generating diverse variations may be better suited to TTT than OPRO, which focuses more on direct improvement of previous solutions.

## C  APPENDIX C: LLM PROMPTS

### C.1  SEMANTLE: TASK PROMPT

```
Your task is to guess a hidden word from the English
dictionary. Stick to proper, single-word English words.
Now, guess exactly n=%s new word(s) that could be the
hidden word. Be creative! (Note: give only a list of word(s)
in the provided JSON format, e.g. "response": ["word1",
"word2",...])
```

### C.2  SEMANTLE: NEIGHBORHOOD SAMPLING PROMPT

```
Your task is to guess words related to a word from the
English dictionary. Stick to proper, single-word English
words. Now, guess exactly n=%s new word(s) that could be
related to the word(s):

Word: %s

Be creative! (Note: give only a list of word(s) in
the provided JSON format, e.g. "response": ["word1",
"word2",...])
```

### C.3  DOCKSTRING: TASK PROMPT

```
Your task is to find the optimal drug molecule that has
both a high druglikeness (QED) as well as a strong binding
affinity (vina) with the protein %s. For docking, lower
is better (less than --10 is considered good) and for
druglikeness, 1 is the best and 0 is the worst (greater
than 0.8 is considered good). While both properties are
important, the docking score is 10 times as important as the
druglikeness score. If you propose an invalid molecule or
make a repeat guess, you will get no score, so stick to valid
SMILES strings.

Now, guess exactly n=%s new molecule(s).

(Note: give only a list of SMILES string(s) in the provided
JSON format, e.g. "response": ["SMILES1", "SMILES2", ...])
```

### C.4  DOCKSTRING: NEIGHBORHOOD SAMPLING PROMPT

```
Your task is to find the optimal drug molecule that has
both a high druglikeness (QED) as well as a strong binding
affinity (vina) with the protein %s. For docking, lower
is better (less than --10 is considered good) and for
druglikeness, 1 is the best and 0 is the worst (greater
than 0.8 is considered good). While both properties are
important, the docking score is 10 times as important as the
druglikeness score. If you propose an invalid molecule or
make a repeat guess, you will get no score, so stick to valid
SMILES strings!
```

```
Here is my guess for a molecule:
SMILES: %s

Now, guess exactly n=%s new variation(s) of my molecule that
could improve the scores to reach the optimal molecule.

(Note: give only a list of SMILES string(s) in the provided
JSON format, e.g. "response": ["SMILES1", "SMILES2", ...])
```

## C.5 ARC: Task Prompt

```
Given input-output grid pairs as reference examples,
carefully observe the patterns to predict the output grid
for new test input. Each pair follows the same transformation
rule. Grids are 2D arrays represented as strings, with cells
(colors) separated by spaces and rows by newlines. Here are
the input and output grids for the reference examples:

Example 1:
Input:
[[1,1,1,...,1]]
Output:
[[2,2,2,...,2]]

Example 2:
Input:
[[2,2,2,...,2]]
Output:
[[3,3,3,...,3]]

...

Here is the input grid for the test example:
Input:
[[3,3,3,...,3]]

Write a Python function `transform` that can convert any
given input grid to its corresponding output grid based on
the pattern observed in the reference examples.
```

## C.6 ARC: Neighborhood Sampling Prompt

```
Given input-output grid pairs as reference examples,
carefully observe the patterns to predict the output grid
for new test input. Each pair follows the same transformation
rule. Grids are 2D arrays represented as strings, with cells
(colors) separated by spaces and rows by newlines.

Here are the input and output grids for the reference
examples:

Example 1:
Input:
[[1,1,1,...,1]]
Output:
[[2,2,2,...,2]]

...

Here is the input grid for the test example:
```

```
Input:
[[3,3,3,...,3]]

The goal is to write a Python function `transform` that can
convert any given input grid to its corresponding output
grid based on the pattern observed in the reference examples.

Here is my guess for the function:
```python
def transform(input: np.ndarray) -> np.ndarray:
    # Code
```

Provide a variation of my guess that could be the correct
answer.
```

## C.7 DISCOVERYBENCH: TASK PROMPT

```
You are a research scientist who is interested in data-driven
research using the provided dataset(s) and query. Be creative
and think of an interesting new experiment to help answer
the provided scientific query. Explain in natural language
the experiment plan that the programmer should follow (do not
provide the code yourself). Here are a few instructions that
you must follow:

1. Strictly use only the dataset(s) provided and do not
simulate dummy/synthetic data or columns that cannot be
derived from the existing columns.

2. The experiment plan should be creative, independent, and
self-contained.

3. Use the prior experiments (if any) as inspiration to think
of an interesting and creative new experiment. However, do
not repeat the same experiments.

Here is a possible approach to coming up with a new
experiment plan:

1. Find an interesting context: this could be a specific
subset of the data. E.g., if the dataset has multiple
categorical variables, you could split the data based on
specific values of such variables, which would then allow
you to validate a hypothesis in the specific contexts defined
by the values of those variables.

2. Find interesting variables: these could be the columns
in the dataset that you find interesting or relevant to the
context. You are allowed and encouraged to create composite
variables derived from the existing variables.

3. Find interesting relationships: these are interactions
between the variables that you find interesting or relevant
to the context. You are encouraged to propose experiments
involving complex predictive or causal models.

4. You must require that your proposed experiment plan is
based on robust statistical tests. Remember, your programmer
can install python packages via pip which can allow it to
write code for complex statistical analyses.
```

```
5. Multiple datasets: If you are provided with more than one
dataset, then try to also propose an experiment that utilize
contexts, variables, and relationships across datasets, e.g.,
this may involve using join or similar operations.

"Generally, in typical data-driven research, you will need
to explore and visualize the data for possible high-level
insights, clean, transform, or derive new variables from the
dataset to be suited for the investigation, deep-dive into
specific parts of the data for fine-grained analysis, perform
data modeling, and run statistical tests.

Examples of valid experiment plans:

Experiment plan #1:

1. Merge the datasets offshore, immigration, and
native_employment on the common columns 'year' and 'beaind'.

2. Replace infinite values with NaNs and drop rows with NaNs
in any column.

3. Independent variables: 'iv_offshoring_1', 'penetration'

4. Fit the OLS regression modela

Experiment plan #2:

1. Chose BMI as dependent variable.

2. Time preference (independent) variables as 'DISSAVED' and
'SAMESAVE'.

3. Fit an OLS regression model and returned the model
summary.

Plan an experiment to answer the question about the following
dataset.

{dataset_metadata}

Now create exactly {n} new experiment plans that could
answer the scientific question. Note: give only a list
of experiment plans in the provided JSON format, e.g.
{"response": ["experiment_plan_1", "experiment_plan_2", ...]})
```

## C.8 DISCOVERYBENCH: NEIGHBORHOOD SAMPLING PROMPT

```
You are a research scientist who is interested in data-driven
research using the provided dataset(s) and query. Be creative
and think of an interesting new experiment to help answer
the provided scientific query. Explain in natural language
the experiment plan that the programmer should follow (do not
provide the code yourself). Here are a few instructions that
you must follow:

1. Strictly use only the dataset(s) provided and do not
simulate dummy/synthetic data or columns that cannot be
derived from the existing columns.

2. The experiment plan should be creative, independent, and
self-contained.
```

3. Use the prior experiments (if any) as inspiration to think of an interesting and creative new experiment. However, do not repeat the same experiments.

Here is a possible approach to coming up with a new experiment plan:

1. Find an interesting context: this could be a specific subset of the data. E.g., if the dataset has multiple categorical variables, you could split the data based on specific values of such variables, which would then allow you to validate a hypothesis in the specific contexts defined by the values of those variables.

2. Find interesting variables: these could be the columns in the dataset that you find interesting or relevant to the context. You are allowed and encouraged to create composite variables derived from the existing variables.

3. Find interesting relationships: these are interactions between the variables that you find interesting or relevant to the context. You are encouraged to propose experiments involving complex predictive or causal models.

4. You must require that your proposed experiment plan is based on robust statistical tests. Remember, your programmer can install python packages via pip which can allow it to write code for complex statistical analyses.

5. Multiple datasets: If you are provided with more than one dataset, then try to also propose an experiment that utilize contexts, variables, and relationships across datasets, e.g., this may involve using join or similar operations.

"Generally, in typical data-driven research, you will need to explore and visualize the data for possible high-level insights, clean, transform, or derive new variables from the dataset to be suited for the investigation, deep-dive into specific parts of the data for fine-grained analysis, perform data modeling, and run statistical tests.

Examples of valid experiment plans:

Experiment plan #1:

1. Merge the datasets offshore, immigration, and native_employment on the common columns 'year' and 'beaind'.

2. Replace infinite values with NaNs and drop rows with NaNs in any column.

3. Independent variables: 'iv_offshoring_1', 'penetration'

4. Fit the OLS regression modela

Experiment plan #2:

1. Chose BMI as dependent variable.

2. Time preference (independent) variables as 'DISSAVED' and 'SAMESAVE'.

3. Fit an OLS regression model and returned the model summary.

Plan an experiment to answer the question about the following dataset.

```
{dataset_metadata}

PRIOR EXPERIMENTS

Now create exactly {n} new experiment plans that could
answer the scientific question and are **similar** to the
prior experiments. Note: give only a list of experiment
plans in the provided JSON format, e.g. {"response":
["experiment_plan_1", "experiment_plan_2", ...]})
```

