# OpenReview forum: "MiGrATe: Mixed-Policy GRPO for Adaptation at Test-Time"
_ICLR.cc/2026/Conference — Submitted to ICLR 2026_

### Official Review · Reviewer_RWAj · 2025-10-27

**Soundness:** 2
**Presentation:** 2
**Contribution:** 2
**Rating:** 2
**Confidence:** 3

**Summary:**

This submission focuses on balancing exploration and exploitation when applying large language models (LLMs) to the black-box optimization tasks. Notably, test-time training (TTT) shows a promising direction by training the model with synthetic data for enhancing the solution quality. However, the data for TTT is limited and requires task-specific design. To address the data feasibility problem, this submission proposes MIGRATE, an online TTT algorithm that employs a mixed search algorithm for constraining data and adopts group relative policy optimization (GRPO) to optimize the policy model. The submission conducts extensive empirical studies and justifies the effectiveness of MIGRATE.

**Strengths:**

- The submission focuses on the trendy language model reasoning problem, identifying the weakness of prior TTT methods, and proposes an effective method to handle the data feasibility problem.
- The submission is generally well-written, with clear illustrations and tables.
- The conducted experiments provide empirical evidence on the effectiveness of the proposed method compared to the baselines.

**Weaknesses:**

- The submission claims a focus on the test-time training. However, MIGRATE updates the model's parameters using reinforcement learning and samples with verification. This violates the definition of TTT [1], where the model is trained in a self-supervised manner.
- The core idea of MIGRATE is employing off- and on-policy sampling to improve the data scale and quality, and directly using the sampled data to train the model using GRPO without any further adaptation to the test-time training setting.
- The adopted baselines are primarily prompt-based approaches, such as OPRO and self-reflection. There are no comparisons with other TTT and RL-based approaches. Furthermore, related black-box optimization approaches are also missing, such as evolutionary algorithms or the Bayesian optimization approach. This weakens the reliability of the conducted experiments.

[1] Test-Time Training with Self-Supervision for Generalization under Distribution Shifts. In ICML, 2020.

**Questions:**

1. Could you provide experiments on comparing with the iterative search algorithm, like MCTS or the evolutionary algorithm? How about other TTT methods?
2. Could you provide further discussion or empirical evidence on the "small variations in solution yield small changes in quality" claim in Sec. 4.1, Neighborhood sampling paragraph? Since the datasets like ARC and Dockstring might yield completely different quality of solution with mild modification.

---

> ### Public Comment · ~Yu_Sun1 · 2025-11-13
>
> I am honored and humbled that the reviewer cites my paper [1] for the definition of test-time training, but the idea of test-time training has a long history and should not be defined by a single paper, which is why I chose the name "Test-Time Training with Self-Supervision" to narrow down the scope of my paper. The definition should include any method that formulates a learning problem according to a test instance, and restricting to "where the model is trained in a self-supervised manner" should not be necessary.

---

> ### Author Response · Authors · 2025-11-25
>
> > `Could you provide experiments on comparing with the iterative search algorithm, like MCTS or the evolutionary algorithm? How about other TTT methods?`
>
> In addition to the existing set of iterative search baselines (Random sampling, Neighbourhood-sampling (NS), OPRO, GRPO, and GRPO-Greedy), we have now also added (1) an evolutionary algorithm (Evolution) and (2) a second RL algorithm (online DPO), for the tasks of Semantle, Dockstring, and ARC-small (we will expand this to ARC-full and DiscoveryBench in our camera-ready version). We are also working on adding a Bayesian optimization baseline, BOPRO [3], which we hope to provide results for in the next few days.
>
> Evolution refers to using an island-based evolutionary strategy to select in-context examples for neighborhood sampling. This is similar to prompting for full rewrites with inspirations provided in-context, which is seen in related works such as AlphaEvolve[4] and OpenEvolve [2]. Our earlier version also included this baseline (Appendix B1), but only for a small assessment. We have now expanded it to our main results.
> Online DPO [5] samples on-policy completions in each iteration, which are used to construct preference pairs and calculate a policy gradient using the standard DPO objective.
> BOPRO [3] uses latent space Bayesian optimization over solution embeddings to search for better sampling distributions via context engineering over past solutions.
>
> Our results (updated in Tables 1 and 6) show that Evolution performs similarly to our existing baseline of OPRO (though underperforming it on ARC-small), while online DPO performs much worse than our existing RL baselines (GRPO and its variants). We additionally found search with online DPO harder to tune, resulting in more inadmissible solutions (i.e., invalid sequence of tokens) as compared to other RL baselines.
>
> ---
>
> > `Could you provide further discussion or empirical evidence on the "small variations in solution yield small changes in quality" claim in Sec. 4.1, Neighborhood sampling paragraph?  Since the datasets like ARC and Dockstring might yield completely different quality of solution with mild modification.`
>
> Thank you for this question! Our *continuity assumption* refers to small changes in the **parameter space** correlating with small changes in solution quality. Our earlier text was indeed ambiguous, and we have updated this in the new version. Additionally, we provide a small visualization (in Figure 2) empirically demonstrating this effect. Specifically, we compute pairwise L2 norms between LoRA weights across 100 training iterations using MiGrATe and plot this distance against the difference in average quality of sampled solutions using those weights. As shown, at low L2 norms, the difference in quality is very low, while at higher L2 norms, the difference in quality may be both low or high, which is expected when taking large jumps in the parameter space during search.
>
> ---
>
> > `However, MIGRATE updates the model's parameters using reinforcement learning and samples with verification. This violates the definition of TTT [1], where the model is trained in a self-supervised manner.`
>
> As described by Yu Sun (author of [1]) very kindly (thank you!), our method falls within the broader definition of test-time training (TTT), which involves updating a model's parameters using only the test instance. In MiGrATe, the "training" (or parameter updates) refers to iteratively searching for a new set of LoRA parameters, which can result in generating higher-performing solutions for the test instance.
>
> ---
>
> **References**
>
> [1] The Surprising Effectiveness of Test-Time Training for Few-Shot Learning (Akyürek, 2024)
> [2] OpenEvolve: an open-source evolutionary coding agent (Sharma 2025)
> [3] Searching for Optimal Solutions with {LLM}s via Bayesian Optimization (Agarwal et al., 2025)
> [4] AlphaEvolve: A coding agent for scientific and algorithmic discovery (Novikov et al., 2025)
> [5] Direct Language Model Alignment from Online AI Feedback (Guo et al., 2024)

---

> > ### Comment · Reviewer_RWAj · 2025-11-27
> >
> > I thank the authors for their detailed responses and the clarification of TTT from the authors and Yu Sun.
> > The additional experiments and discussion address my concerns about the submission.
> > I have no further questions and raise my score from 2 to 4.

---

### Official Review · Reviewer_vGkA · 2025-10-29

**Soundness:** 3
**Presentation:** 3
**Contribution:** 2
**Rating:** 4
**Confidence:** 3

**Summary:**

This paper proposes a mixed sampling framework for GRPO in LLM test-time training. The authors use completions sampled from historical completion databse and their neighborhoods to enhance exploitation during test-time training. Experimental results shows that the proposed method surpasses inference-only and test-time training baselines in multiple optimization scenarios.

**Strengths:**

1. The introduction of greedy and neighborhood sampling from a historical database reduces reward sparsity in complex optimization scenarios and lowers the expertise required for offline data preparation.

2. The proposed method outperforms GRPO-based, test-time training variants. The authors also provide a detailed analysis of the exploration-exploitation tradeoff in the mixed sampling method.

**Weaknesses:**

1. At the beginning of the solution search with MiGrATe, the solutions in the database might not be high-performing, as might the top-k solutions and the neighborhood solutions derived from them. Since a large proportion of solutions may be derived from the database initially (greedy sampling + neighborhood sampling), MiGrATe's performance might be significantly influenced by the initial sampling, which can be variable. This could be problematic when the reward is sparse and on-policy sampling fails to generate solutions that elicit non-zero rewards, the database solutions might mislead the LLM training into converging to local optima.

2. The proposed method requires different $[\alpha, \gamma, \beta]$ ratios for different tasks (e.g., [0, 4, 1] for Semantle and [2, 2, 1] for Dockstring), which are also carefully hand-crafted and task-specific. This introduces additional expertise requirements or necessitates preliminary experiments to set proper values, which might limit the method's generality and hinder further application to other optimization tasks without prior knowledge.

**Questions:**

1. At the beginning of the solution search with MiGrATe, the database is empty. How to sample the top-k solutions?

2. In the MiGrATe hyperparameters for Semantle, on-policy samples $\alpha$ is 0, then how to fill database without on-policy sampling?

---

> ### Author Response · Authors · 2025-11-25
>
> > `At the beginning of the solution search with MiGrATe, the solutions in the database might not be high-performing, as might the top-k solutions and the neighborhood solutions derived from them.`
>
> > `At the beginning of the solution search with MiGrATe, the database is empty. How to sample the top-k solutions?`
>
> This is an excellent point and refers to the common "cold-start" problem in black-box optimization settings, where the prior distribution may not be informative, leading to a larger search budget to find good-quality solutions. While MiGrATe is not immune to this problem, the use of an LLM as the generative model promises to provide a broad prior, which can then be used to generate a diverse set of warmstart / space-filling candidates to bootstrap search.
>
> For Semantle specifically, we initialize the database with 20 diverse warmstart words. In the case of other domains where there is not a provided warmstart, we generate a warmstart set using the initial proposal distribution of the LLM, i.e., the base model without modifying any of its parameters.
>
> Nonetheless, we do agree that an LLM may have poor priors in various domains, and this makes for important future work to conduct, e.g., to efficiently incorporate human input, or induce better exploration abilities within models.
>
> ---
>
> > `The proposed method requires different  ratios for different tasks, which are also carefully hand-crafted and task-specific.`
>
> We acknowledge that our method requires tuning the $\alpha$, $\beta$, and $\gamma$ parameters based on the search domain (sensitivity analysis presented in Appendix B.3). We draw a connection between the tuning required in MiGrATe with the tuning of priors in Bayesian optimization, where knowledge of the domain helps inform the hyperparameters of the kernel function. We also note that our evolutionary computation baseline Evolution also comes with several hyperparameters (e.g., number of islands, population size, archive size, migration interval, migration rate, elite selection ratio, exploration ratio, exploitation ratio, etc.), which must be tuned in order to see good performance.
>
> ---
>
> > `In the MiGrATe hyperparameters for Semantle, on-policy samples  is 0, then how to fill database without on-policy sampling?`
>
> Even if on-policy samples are set to 0, we use the warmstarting procedure described above to initialize the database, following which neighborhood and greedy sampling is performed.

---

### Official Review · Reviewer_WbxR · 2025-10-30

**Soundness:** 2
**Presentation:** 2
**Contribution:** 2
**Rating:** 2
**Confidence:** 3

**Summary:**

This paper explores how to test-time-train (TTT) base LLMs for optimization tasks. Started from OPRO (an initial google's llm for optimization Q&A framework), the authors listed weaknesses of such in-context learning (e.g., exploration-exploitation tradeoff inability) and notive themselves  to address this issue by introducing TTT concepts. Specifically, the authors consider fine-tuning LLMs with self-generated  data. This is achieved by replacing the naive grouping data in GRPO with three parts of self-generated data: 1) on-policy sampling data, same as standard policy gradient methods; 2) on-policy historical demonstrations, a database is used to store historical samples generated before current training step, and topk samples are selected; 3) local exploration, by prompting LLMs to generate small. stocastic variations on the elite historical demonstrations. Using these data, the LLMs are fine-tuned via balanced exploration and exploitation to achieve better solutions on tested problems. The authors validate this approach's effectiveness through comparing the trained LLMs (with LoRA) with OPRO and pther specialized baselines on diverse optmization and reasoning tasks.

**Strengths:**

1. I think this paper presents a novel perspective for LLMs as Optimizers works such as OPRO. The authors locate the exploration and exploitation imbalance in such in-context learning approaches and make an interesting try on using TTT to rebalance such tradeoff.

2. I appreciate the authors provide the code for reproducibility checking.

**Weaknesses:**

1. While I acknowedge that the overall methodology the authors have proposed are solid and interesting (self-supervision), I have to say that  I can not see real and practical value of this work for real-world optimization problems. I can understand that the authors may not be long-standing optimization researchers, however, in realistic scenario, using the method you provide may not be practical since it requires training LLMs for solving one problem. I found this is quite opposite to existing automated algorithm design or learning to optimize or meta-black-box optimization, where the aim is to learn optimzers that generalize across different problems. Based on this point, I can not support this paper as a useful optimization approach.

2. I think the writing of this paper is not clear at all. First, I can not understand what the term "optimization" denotes until I  read section 5 (experiment). Even if I read section 5, I still can not understand what is the optimizer in this paper, is it the LLM itself? Second, What is the relationship between the related works (EC, RLVR) and this paper? I can not understand why the authors mention these works since you never compare them in experiments. Third, the NS sampling is not explained with clarity (lines 220 - xx). About the NS sampling. another question is why "continuity assumption" persists in your test cases, for LLMs, an intuitive feeling is that once you slightly change the prompt, the output may vary significantly, and for the testing scenarios, a small change in the solution also can not ensure small chance in its evaluation score, such as ARC.

3. Objectively speaking, OPRO is not a qualified optimizer at all, as validated in recent papers.  Besides, OPRO has no actual training, hence it is somewhat unfair to compare your mothod with it.

4. For reasoning task such as ARC, please also provide performance of existing strong LLMs such as GPT-5, otherwise I can not tell if your method is really a good and useful one.

**Questions:**

See Weaknesses.

---

> ### Author Response · Authors · 2025-11-25
> **Response (1/2)**
>
> > `What is the relationship between the related works (EC, RLVR) and this paper?`
>
> Evolutionary computation (EC) represents a canonical family of methods to solve problems in the BBO setting in which we operate. In our new version, we now expand our small-scale evaluation of EC to our main results:
> - Table 1:
>   - Evolution (Semantle, Dockstring)
> - Table 6:
>   - Evolution (ARC-Small)
>   - MiGrATe (\gamma-Evolution) (ARC-Small)
>
> Evolution refers to using an island-based evolutionary strategy to select in-context examples for neighborhood sampling. This is similar to prompting for full rewrites with inspirations provided in-context, which is seen in related works such as AlphaEvolve[3] and OpenEvolve [1]. We will expand our evaluations to ARC-full and DiscoveryBench in our camera-ready version.
>
> As for RLVR, the main connection we were hoping to draw was the use of static, scalar reward functions in tandem with RL fine-tuning. But we agree that this is a rather broad connection, which is better drawn with the BBO setting, more generally. In lieu of this, we have now added a new paragraph describing connections between this work and Bayesian optimization within the context of LLMs, in particular.
>
> ---
>
> > `About the NS sampling. another question is why "continuity assumption" persists in your test cases…once you slightly change the prompt, the output may vary significantly…small change in the solution also can not ensure small chance in its evaluation score…`
>
> We apologize for the confusion caused by our earlier writing. Our *continuity assumption* refers to small changes in the **parameter space** correlating with small changes in solution quality. We have updated this in the new version. Additionally, we provide a small visualization (in Figure 2) empirically demonstrating this effect. Specifically, we compute pairwise L2 norms between LoRA weights across 100 training iterations using MiGrATe and plot this distance against the difference in average quality of sampled solutions using those weights. As shown, at low L2 norms, the difference in quality is very low, while at higher L2 norms, the difference in quality may be both low or high, which is expected when taking large jumps in the parameter space during search.
>
> ---
>
> > `…OPRO is not a qualified optimizer…OPRO has no actual training, hence it is somewhat unfair to compare your method with it.`
>
> Each search algorithm we consider in our work operates by modifying in each iteration (over a fixed budget) the sampling distribution of the LLM either by making changes to the input to the LLM or by varying a subset of its parameters. OPRO takes the former approach and optimizes the input prompt by using an algorithm that selects from the set of all past solutions it has generated so far. On the other hand, in MiGrATe, we perform lightweight gradient updates to a small set of LoRA parameters, which in turn modifies the sampling distribution to iteratively generate better solutions (again, in the same fixed budget as OPRO). This is to say that none of our test-time training baselines (including MiGrATe) utilize signal not available to inference-only baselines (e.g., OPRO).
>
> Furthermore, we provide runtime details in Appendix A.3 comparing TTT-based MiGrATe with an inference-only baseline, highlighting that MiGrATe does not require significantly higher computational resources. Across baselines, we, in fact, find that the primary source of latency and resource requirements come from the solution sampling and the task-specific black-box verifiers. Since the budget for these generations and black-box computations are consistent across methods (meaning all methods propose the same number of solutions), we argue that our baseline comparisons are indeed fair.
>
> ---
>
> >  `I can not understand what the term "optimization" denotes until I read section 5 (experiment)`
>
> Thank you for this feedback! We clarify that "optimization" in our work refers to black-box search over the space of all possible LoRA parameters, which in turn modifies the LLM's sampling distribution to generate task and problem instance-specific solution candidates (e.g., words, molecules, programs, hypotheses) using a reward function. The "optimizer" in this work is the MiGrATe algorithm, which changes the set of LoRA parameters in each iteration, which can be seen as akin to standard optimizers like Adam which define how parameters are modified. We have added a new line (L78) to our introduction to make this more clear. Furthermore, we have added a paragraph to our Related Work section on Bayesian optimization to also ground our use of "optimization" within BO terminology. We hope that this addresses your concern.

---

> > ### Author Response · Authors · 2025-11-25
> > **Response (2/2)**
> >
> > > `…using the method you provide may not be practical since it requires training LLMs for solving one problem…opposite to existing automated algorithm design…where the aim is to learn optimizers that generalize across different problems…`
> >
> > Our main goal in this work is to combine research in automated *experiment* design with generative capabilities of large language models, which provide broad knowledge (from pre-training) and reasoning (from post-training), to tackle hard natural-language based search problems such as molecule discovery and hypothesis generation (as also being tackled by contemporary methods such as AlphaEvolve). Our focus here is on the BBO [5, 6, 7] setting only, where the learned surrogate may be discarded after search. We, however, do acknowledge that the setting of meta-BBO, which aims to generalize across problem instances, makes for very interesting future work. In fact, we present a small evaluation in Appendix B.2 (Figure 7) to evaluate whether optimized parameters for one problem instance can help bootstrap optimization for another instance.
> >
> > Given our "single-use" setting, we indeed agree that learning efficiency is important to consider. In MiGrATe, we, therefore, optimize only a lightweight subset of LoRA parameters added on to a pre-trained LLM, which provides practical feasibility to perform fast gradient-based updates (please see runtimes in Appendix A.3). Borrowing terminology from BBO, the surrogate model in our setting is the LLM itself and MiGrATe may be considered as the acquisition function to decide which point to sample from next.
> >
> > ---
> >
> > >  `…provide performance of existing strong LLMs such as GPT-5...`
> >
> > All results presented in our work within each search task use the same underlying LLM as the backbone, allowing for clear comparisons between different search algorithms. In particular, we use `Llama-3.1-ARC-Potpourri-Induction-8B` for the program synthesis task of ARC, `Llama-3.2-3B-Instruct` for Semantle, `Llama-3.2-3B-Instruct` for Dockstring, and `Qwen2.5-7B-Instruct` for DiscoveryBench.
> > We emphasize that the goal of our work is to show that problem instance-specific parameter updates is a practical and performant methodology to search for natural-language solutions. We expect our findings to transfer over to other model families (e.g., OpenAI, provided there exists fine-tuning access).
> >
> > ---
> > **References:**
> >
> > [1] The Surprising Effectiveness of Test-Time Training for Few-Shot Learning (Akyürek, 2024)
> > [2] AlphaEvolve: A coding agent for scientific and algorithmic discovery (Novikov et al., 2025)
> > [3] Combining Induction and Transduction for Abstract Reasoning (Li et al., 2024)
> > [4] OpenEvolve: an open-source evolutionary coding agent (Sharma 2025)
> > [5] A Tutorial on Bayesian Optimization (Frazier, 2018)
> > [6] A New Method of Locating the Maximum Point of an Arbitrary Multipeak Curve in the Presence of Noise (Kushner, 1964)
> > [7] Efficient Global Optimization of Expensive Black-Box Functions (Jones et al., 1998)

---

### Official Review · Reviewer_Z37p · 2025-11-01

**Soundness:** 3
**Presentation:** 3
**Contribution:** 3
**Rating:** 8
**Confidence:** 4

**Summary:**

The paper presents a method for online test-time training of large language models that enables adaptive search in black-box optimization tasks without requiring external training data. The approach uses GRPO with a novel mixed-policy group construction strategy that combines on-policy sampling to ensure exploration by sampling from the current policy, greedy sampling to exploit known high-reward regions by reusing top-performing past completions, and neighborhood sampling to facilitate local exploration by generating structural variations of high-reward solutions. The method iteratively adapts LLM parameters at test time to shift the sampling distribution toward higher-quality solutions. The authors test their approach across four domains including, showing improvements over both inference-only and test-time training baselines.

**Strengths:**

The paper's most significant contribution is its generalizable approach that eliminates the need for handcrafted, task-specific training data, which has been a major limitation of prior test-time training methods. All training signals in MIGRATE are model-generated, making the method applicable across diverse domains without requiring domain expertise to curate training examples, particularly helpful for low data domains.

The design of the three-component mixed-policy strategy is well-motivated and balances the exploration-exploitation tradeoff. This combination addresses fundamental challenges in black-box optimization that purely on-policy or off-policy methods struggle with individually.

The experimental validation is comprehensive and strengthens confidence in the method. The authors evaluate on diverse tasks with different solution spaces and reward functions, compare against multiple baselines including both inference-only methods (Random, NS, OPRO, Reflexion) and test-time training variants (GRPO, GRPO-Greedy), and include extensive ablations showing each component's contribution. The analysis goes beyond simple accuracy metrics to examine search behaviors, solution quality distributions, and trajectory visualizations, providing insights into how and why the method works.

**Weaknesses:**

The computational cost is not thoroughly discussed, which is a significant oversight for a test-time training method. While runtime is mentioned in the appendix (for example, 51 minutes per ARC task on an A100 GPU), there is insufficient analysis comparing the cost versus inference-only methods, examining memory requirements for LoRA fine-tuning, assessing scalability to larger models or longer horizons, or analyzing trade-offs between TTT overhead and solution quality improvements.

The method also shows hyperparameter sensitivity, requiring careful tuning of α, β, and γ for each domain. Figure 7 clearly shows that optimal ratios vary significantly across tasks. A more clear or principled guide on these parameters would be very helpful for broad usability.

The paper does not provide formal analysis of why this particular mixed-policy strategy works, no validation of the "continuity assumption" underlying neighborhood sampling beyond empirical results, no convergence guarantees or sample complexity bounds, and no clear characterization of when or why the method might fail. While empirical validation is valuable, theoretical understanding would provide confidence about generalization to new domains and guidance for method development.

The improvements on some tasks are actually quite small, so discussing where the method best works and why would be valuable.

Several evaluation limitations discussed by the authors affect the conclusiveness of the results (such as ARC 200/400 tsks evaluated, single runs with boostraping for variance estimates).

The implementation details of neighborhood sampling remain somewhat unclear. The paper doesn't fully explain how "stochastic variations" are generated from the prompt, what ensures they remain in the neighborhood versus being arbitrary perturbations, or how the quality of NS depends on prompt engineering. This makes the method harder to reproduce and potentially sensitive to prompt formulation in ways that aren't explored.

The comparison to related work is incomplete in several respects. Evolutionary algorithms are only briefly compared in the appendix, there's no comparison to Bayesian optimization methods despite these being mentioned in related work, and comparisons to other recent TTT methods beyond GRPO variants are missing. Broader comparisons would strengthen the claims.

**Questions:**

Can the authors provide formal analysis or intuition about when MIGRATE is expected to outperform pure on-policy or pure off-policy approaches? Are there identifiable task characteristics that predict method effectiveness, such as properties of the reward landscape, solution space structure, or base model capabilities?

Second, the hyperparameter selection process needs clarification. How should someone choose α, β, and γ for new tasks where the optimal configuration is unknown?

Can the authors provide more technical details on how NS actually generates variations? Have they tried alternative NS implementations, such as explicit perturbations in latent space rather than prompt-based variation? How sensitive is performance to the specific NS prompt formulation, and what makes a good NS prompt?

Understanding failure modes would provide important context. Are there scenarios where MIGRATE performs worse than baselines, perhaps due to misleading initial solutions or particular reward landscape properties? What happens in extremely sparse reward settings where even greedy samples have near-zero reward? How does the method behave when the base model is very weak versus very strong?

---

> ### Author Response · Authors · 2025-11-25
> **Response (1/2)**
>
> > `Evolutionary algorithms are only briefly compared in the appendix, there's no comparison to Bayesian optimization methods despite these being mentioned in related work, and comparisons to other recent TTT methods beyond GRPO variants are missing.`
>
> In addition to the existing set of iterative search baselines (Random sampling, Neighbourhood-sampling (NS), OPRO, GRPO, and GRPO-Greedy), we have now also added (1) an evolutionary algorithm (Evolution) and (2) a second RL algorithm (online DPO), for the tasks of Semantle, Dockstring, and ARC-small (we will expand this to ARC-full and DiscoveryBench in our camera-ready version). We are also working on adding a Bayesian optimization baseline, BOPRO [1], which we hope to provide results for in the next few days.
>
> Evolution refers to using an island-based evolutionary strategy to select in-context examples for neighborhood sampling. This is similar to prompting for full rewrites with inspirations provided in-context, which is seen in related works such as AlphaEvolve[2] and OpenEvolve [3]. Our earlier version also included this baseline (Appendix B1), but only for a small assessment. We have now expanded it to our main results.
> Online DPO [4] samples on-policy completions in each iteration, which are used to construct preference pairs and calculate a policy gradient using the standard DPO objective.
> BOPRO [1] uses latent space Bayesian optimization over solution embeddings to search for better sampling distributions via context engineering over past solutions.
>
> Our results (updated in Tables 1 and 6) show that Evolution performs similarly to our existing baseline of OPRO (though underperforming it on ARC-small), while online DPO performs much worse than our existing RL baselines (GRPO and its variants). We additionally found search with online DPO harder to tune, resulting in more inadmissible solutions (i.e., invalid sequence of tokens) as compared to other RL baselines.
>
> ---
>
> > `…no validation of the "continuity assumption"...`
>
> Our *continuity assumption* refers to small changes in the **parameter space** correlating with small changes in solution quality. We have updated this in the new version. Additionally, we provide a small visualization (in Figure 2) empirically demonstrating this effect. Specifically, we compute pairwise L2 norms between LoRA weights across 100 training iterations using MiGrATe and plot this distance against the difference in average quality of sampled solutions using those weights. As shown, at low L2 norms, the difference in quality is very low, while at higher L2 norms, the difference in quality may be both low or high, which is expected when taking large jumps in the parameter space during search.
>
> ---
>
> > `A more clear or principled guide on these parameters would be very helpful for broad usability… Can the authors provide formal analysis or intuition about when MIGRATE is expected to outperform pure on-policy or pure off-policy approaches?`
>
> Some general rules of thumbs that we observed across different experimentation settings:
> A pure on-policy configuration with \alpha=N performs inefficiently (this configuration is equivalent to vanilla GRPO in our results). This configuration wastes many iterations by generating the same solutions. This problem is more prominent in a task such as Semantle where low performing guesses provide little signal on what the target word could be (see Fig. 8a). Therefore it becomes more helpful to mandate new samples be from a higher performing solution distribution via NS.
> The configuration of \beta=N performs suboptimally (see Fig. 9). This causes the model to quickly converge to and get stuck in a local optima. This behavior was most evident on ARC where no improvement is observed beyond the first few samples.
>
> Please see Appendix B.3 for a more detailed analysis of the hyperparameters in MiGrATe.
>
> ---
>
> > `The computational cost is not thoroughly discussed...`
>
> Please see Appendix A.3, where we provide information on the computational resources needed by MiGrATe in comparison to inference-only baselines. Across tasks, we find that MiGrATe does not add a significant overhead in latency. We find that the primary source of latency and resource requirements come from solution sampling and task-specific black-box reward functions. The budget for these computations are consistent across search baselines, which is why there is not a significant difference between their runtimes.

---

> > ### Author Response · Authors · 2025-11-25
> > **Response (2/2)**
> >
> > > `The implementation details of neighborhood sampling remain somewhat unclear…`
> >
> > > `…alternative NS implementations, such as explicit perturbations in latent space rather than prompt-based variation?...How sensitive is performance to the specific NS prompt formulation, and what makes a good NS prompt?`
> >
> > Neighborhood sampling (NS) may be viewed as an exploratory variant of OPRO. While in OPRO, the instruction is to extrapolate and improve upon the provided solutions according to their performance metric, NS differs by specifically querying for *variations* of provided solution(s) (without performance metrics). This open-ended approach favors GRPO by generating more diverse samples for better training signal. This result can be seen in Table 6 where MiGrATe with NS consistently outperforms MiGrATe with the OPRO prompt instead. Additionally, please see Appendix C for the exact prompts we use for NS.
> >
> > In general, we find that performance is more sensitive to what examples are chosen to be provided in-context than it is to the specific instruction formulation. For instance, in Semantle, we find that randomly selecting among the top-3 formally generated solutions performs better than always selecting the best (top-1) solution. This is a trend similarly seen in OPRO, where providing the top-10 solutions in context performs better than top-1.
> >
> > Regarding alternative NS implementations, direct perturbations to LoRA parameters is an interesting idea, but it is unclear whether random perturbations would be able to substantially change the sampling distribution in meaningful ways to retain the model’s ability to generate valid solutions. A more informed strategy may be more appropriate as in [5]. On the other hand, in MiGrATe, we use the LLM internal knowledge and reasoning to generate valid perturbations of known past solutions.
> >
> > ---
> >
> > > `What happens in extremely sparse reward settings where even greedy samples have near-zero reward? How does the method behave when the base model is very weak versus very strong?`
> >
> > We added an experiment where we compared MiGrATe and OPRO with varying levels of reward sparsity (Section B.4, Figure 10). As expected, the performance of MiGrATe and OPRO decreases as the sparsity of the rewards increase. In particular, when we use a binary reward (where the only non-zero reward is the optimal solution), we see a large dip in performance in both methods. Interestingly, however, we observe that MiGrATe scales better with increasing reward density than OPRO. In addition, while using the second to highest sparsity setting, MiGrATe was able to show competitive results to the best performance OPRO was able to achieve across any sparsity levels. These results further imply the importance of needing a strong base model to effectively leverage reward signals provided by the respective black-box functions.
> >
> > ---
> >
> > **References:**
> >
> > [1] Searching for Optimal Solutions with {LLM}s via Bayesian Optimization (Agarwal et al., 2025)
> > [2] AlphaEvolve: A coding agent for scientific and algorithmic discovery (Novikov et al., 2025)
> > [3] OpenEvolve: an open-source evolutionary coding agent (Sharma 2025)
> > [4] Direct Language Model Alignment from Online AI Feedback (Guo et al., 2024)
> > [5] Evolution Strategies at Scale: LLM Fine-Tuning Beyond Reinforcement Learning (Qiu et al., 2025)

---

### Author Response · Authors · 2025-11-25
**New Revision**

We sincerely thank all the reviewers for taking time to provide thoughtful and constructive feedback. We also appreciate the positive reception, including finding MiGrATe to be generalizable and removing reliance on external training data (reviewers Z37p, vGkA, RWAj), to address the exploration-exploitation problem in LLM-based search (reviewers Z37p, WbxR, vGkA), and be backed by strong empirical validation (reviewers Z37p, RWAj).

---

We have uploaded a new revision and made the following changes to address all feedback and questions:
- **Evolutionary Methods:** Added Evolution and MiGrATe + Evolution baseline results for Semantle, Dockstring, and ARC-Small (inquired by Z37p, WbxR, and RWAj).
    - On ARC-Small, Evolution achieves worse results than the best inference-only method (OPRO). We also incorporated Evolution into MiGrATe as an alternative to NS sampling and find that it outperforms the OPRO strategy but not our proposed MiGrATe with NS (Table 6).
    - On Semantle, Evolution outperforms other inference-only methods, but has middle-of-the-pack performance on Dockstring (Table 1).
- **Other RL methods:** Added Online DPO on Semantle and Dockstring (inquired by Z37p and RWAj).
    - Online DPO performs worse than GRPO.
    - Implementation details for Online DPO are in Appendix A.1.
- **Bayesian optimization** methods (BOPRO)
    - We are actively working on getting results for BOPRO before discussion period ends.
- **Impact of Reward Sparsity:** Compared MiGrATe and OPRO on varying levels of sparsity (inquired by Z37p, vGkA).
    - Both MiGrATe and OPRO show downward trends in performance as sparsity increases. Notably, MiGrATe is still able to outperform OPRO's best performance while using more sparse rewards (Fig. 9). Section B.4 has more details on our experiment methodology.
- **Continuity Assumption:** Clarified the writing and added a visualization (inquired by Z37p, WbxR, RWAj).
    - Edited Section 4.1 to better reflect our claim. The key point is that small changes in the *parameter space* correlate with small changes in solution quality.
    - New visualization plotting pairwise comparisons of LoRA parameter sets in terms of their distance (x-axis) and average solution quality (y-axis).
- **Latency Comparison** (inquired by Z37p).
    - Added additional latency information for different methods (Section A.3).
    - Added a plot showcasing the time it took for different methods to find its best solution. MiGrATe and OPRO converge to a solution much faster than other methods and MiGrATe does so with a higher success rate.
- Added a section on **Bayesian Optimization with LLMs** to the related works (inquired by WbxR).
    - We removed the RLVR section in the related works.
- Updated the caption in Figure 5 to better explain the results and the advantage MiGrATe showcases over an inference-only method.

---

We would be very happy to follow-up on any additional questions or comments.

---

> ### Author Response · Authors · 2025-12-04
>
> **Bayesian optimization methods (BOPRO) Results:** Added the BOPRO results which were inquired by reviewers Z37p and vGkA.
> - Results are presented in Table 1 for Semantle and Dockstring. BOPRO was also run on a subset of the ARC benchmark (ARC-Small), with the corresponding results shown in Table 6.
> - BOPRO underperforms MiGrATe on Dockstring and all other methods on ARC, and only outperforms MiGrATe on Semantle.

---

### Meta-Review · Area_Chair_KaDT · 2026-01-04

**Summary:**

This paper proposes the method MiGrATe for online test-time training to improve the solution quality from LLMs for black-box function optimization tasks.

A core concern from most reviewers is about the limited baselines, and the authors added “Evolution”, BOPRO and Online DPO, which I believe have strengthened the comprehensiveness of the baselines. There, however, remain some potential concerns: (1) Some baseline results are not available in Table 1 (for ARC and DiscoveryBench) and most new baselines are not shown not shown in the figures. (2) MIGRATE does not quite perform significantly better than the baselines. For example, in Table 1, MIGRATE only outperformed baselines for 3 out of 6 metrics, and the improvement can be quite small (e.g., from -10.28 to -11, from 0.77 to 0.79). Once the authors add the new baselines to Figure 3, we might observe less improvement. (3) It is unclear whether the approach also works for larger scale backbone LLMs since the ones used in the experiments are 3B to 8B models. (4) It is still unclear which method “Evolution” is and whether it is a proper implementation of the best available approach in the family of evolutionary algorithm-based BBO approaches. (5) Reviewer WbxR asked about the performance of “existing strong LLMs such as GPT-5” but the authors seemed to have misunderstood. It’d be good to show the performance of the baselines (that do not require fine-tune API) with stronger LLMs as backbones. (6) Failure mode analyses are missing.

Another shared concern is the meaning and validation of the “continuity assumption” (Reviewer Z37p, Reviewer WbxR). The authors clarified what the “continuity assumption” is for, but it is still a bit unclear whether it is valid. The authors only provided a visualization for one example, without actual systematic validations of the assumption. In Figure 2, the y axis is “avg. reward difference”. It is unclear why this value captures the solution quality.

The reviewers also pointed out issues regarding the clarity of the method description. Reviewer Z37p, Reviewer WbxR raised questions about NS sampling and the authors clarified in the rebuttal.

Regarding the questions about hyperparameter sensitivity and lack of principled guidelines for choosing them (Reviewer vGkA, Reviewer Z37p), the authors provided some intuitions from their observations but did not provide principled guidelines. An argument was made about the fact that Bayesian optimization also has hyperparameters to tune, but it is worth pointing out that there has been a substantial amount of work on meta Bayesian optimization and how transfer learning can set those hyperparameters in a principled way.

The paper can be further strengthened by fully incorporating the baseline results, improving the clarity and addressing the hyperparameter issues together with other outstanding concerns. Given that the paper needs significant updates to address those issues, I believe that another round of review is necessary. Hence I recommend rejection.

**Reviewer Concerns:**

See above for shared concerns.

Reviewer Z37p mentioned concerns about insufficient analyses on computational cost, memory requirements, scalability and tradeoffs between TTT overhead and solution quality. The authors addressed these in Appendix A.3.

Outstanding concerns from Reviewer Z37p include (1) lack of formal analysis, convergence guarantees, sample complexity bounds, failure modes; (2) small improvements on some tasks and evaluation limitations that affect the conclusiveness of the results. The authors did not provide formal analyses, convergence guarantees, bounds or failure mode analyses. The current rebuttal does not seem to directly address (2).

Reviewer WbxR raised concerns about practicality of the work, lack of generalization and usefulness, clarity on what “optimization/optimizer” means, relations with related work, unfair comparison with methods that do not require training. The authors provided some explanations, clarifications, but one outstanding concern is the practicality, which was not clearly addressed in the rebuttal. The authors mentioned that a new paragraph is added related to RLVR but it doesn’t exist in the current version.

Reviewer RWAj raised a concern about the definition of TTT but this was a misunderstanding, and was clarified by an author of a TTT paper cited by this reviewer.

Reviewer vGkA had questions about the details about the approach and the authors addressed those.

**Reviewer Scores:**

Reviewer RWAj mentioned they would raise score to 4 after reading the rebuttal.

For other reviewers, given that there are outstanding concerns, I believe they would not have changed their scores if they had been able to participate in the discussions.

---

### Decision · Program_Chairs · 2026-01-26

Reject